# LATEST TASK-SPECIFIC GRAPH NETWORK SIMULATORS

## ABSTRACT

Simulating object deformations is a critical challenge in many scientific domains, with applications ranging from robotics to materials science. Learned Graph Network Simulators (GNSs) are an efficient alternative to traditional mesh-based physics simulators. Their speed and inherent differentiability make them particularly well-suited for inverse design problems such as process optimization. However, these applications typically offer limited available data, making GNSs difficult to use in real-world scenarios. We frame mesh-based simulation as a meta-learning problem and apply conditional Neural Processes to adapt to new simulation scenarios with little data. In addition, we address the problem of error accumulation common in previous step-based methods by combining this approach with movement primitives, allowing efficient predictions of full trajectories. We validate the effectiveness of our approach, called Movement-primitive Meta-MeshGraphNet (M3GN), through a variety of experiments, outperforming state-of-the-art step-based baseline GNSs and step-based meta-learning methods.

## 1 INTRODUCTION

The simulation of complex physical systems is of paramount importance in a wide variety of engineering disciplines, including structural mechanics (Yazid et al., 2009; Zienkiewicz & Taylor, 2005; Stanova et al., 2015), fluid dynamics (Chung, 1978; Zienkiewicz et al., 2013; Connor & Brebbia, 2013), and electromagnetism (Jin, 2015; Polycarpou, 2022; Reddy, 1994). In particular, the simulation of object deformations under external forces is crucial for, e.g., robotic applications (Scheikl et al., 2022; Wang & Zhu, 2023; Linkerhägner et al., 2023). Mesh-based simulations are appealing for such problems due to the computational efficiency and accuracy of the underlying finite element method (Brenner & Scott, 2008; Reddy, 2019). However, the diversity of the problems to be modeled usually necessitates the development of task-specific simulators to accurately capture the relevant physical quantities (Reddy & Gartling, 2010). Such specialized simulators can be slow and cumbersome to use, especially for large-scale simulations (Paszynski, 2016; Hughes et al., 2005).

Thus, data-driven models trained on reference simulations have gained attention as an appealing alternative (Guo et al., 2016; Da Wang et al., 2021; Li et al., 2022). Among them, general-purpose Graph Network Simulators (GNSs) have recently become increasingly popular (Battaglia et al., 2018; Pfaff et al., 2021; Allen et al., 2022b; 2023; Linkerhägner et al., 2023). GNSs encode the simulated system as a graph of interacting entities whose dynamics are predicted using Graph Neural Networks (GNNs) (Bronstein et al., 2021). These models are one to two orders of magnitude faster than classical simulators (Pfaff et al., 2021) while being fully differentiable, which makes them highly effective for, e.g., inverse design problems (Allen et al., 2022b; Xu et al., 2021).

GNSs are typically trained through simple next-step supervision (Battaglia et al., 2018; Pfaff et al., 2021; Allen et al., 2023). During inference, entire trajectories are simulated by iteratively predicting per-node dynamics from an initial system state in an autoregressive manner. This approach is prone to error accumulation over time, especially as the input distribution diverges from the training set (Brandstetter et al., 2022; Han et al., 2022). To mitigate this issue, existing approaches add noise to the input during training and predict the dynamics based on the original inputs, thus following an implicit de-noising objective (Pfaff et al., 2021; Brandstetter et al., 2022). While this significantly improves simulation stability over longer rollouts, the auto-regressive inference of next-step GNSs still propagates and accumulates errors made in earlier prediction steps. These methods also struggle

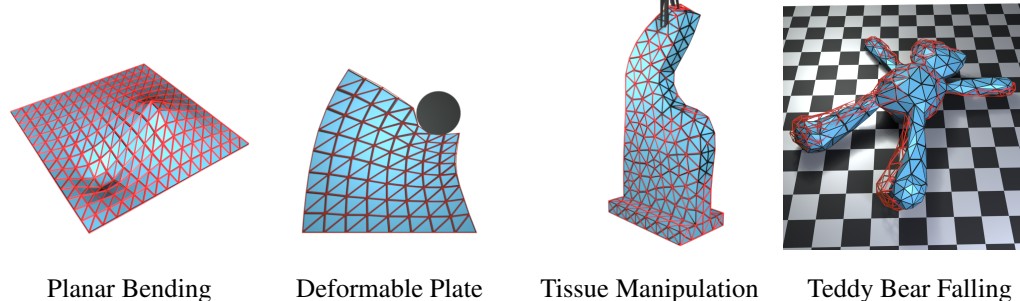

|  |  |  |  |
|---|---|---|---|
| Planar Bending | Deformable Plate | Tissue Manipulation | Teddy Bear Falling |

Figure 1: Final M3GN (Movement-Primitive Meta-MeshGraphNet) simulation steps for different evaluation tasks. From left to right: a planar sheet bending under two orthogonal forces, a falling collider deforming a 2D plate, a surgical tool dragging tissue, and a falling teddy bear. All visualizations present the **predicted mesh** alongside a reference **wireframe** of the ground-truth simulation. M3GN takes the deformable object positions from a few initial time steps and applies meta-learning techniques to infer physical properties, such as Poisson's ratio or Young's modulus. With this latent task description, it predicts the remaining simulation steps using per-node movement primitives.

with partially known initial system states (Linkerhägner et al., 2023), which are common in, e.g., robotic planning (Antonova et al., 2022). Extensions that address this uncertainty (Linkerhägner et al., 2023) require a consistent stream of auxiliary information, such as point cloud observations, throughout the simulation. Furthermore, GNSs typically require large amounts of training data, which prevents their application in real-world scenarios where data is scarce, highlighting the need for efficient adaptation techniques to novel tasks.

To address these limitations, we reformulate learned mesh-based simulation as a trajectory-level meta-learning problem. Here, the initial mesh states of a trajectory serve as a context set on which to condition future predictions. We employ Conditional Neural Processs (CNPs) (Garnelo et al., 2018a) to aggregate these context sets and the dynamics inferred from them into a latent descriptor, which is then used to predict the rest of the trajectory. This method enables efficient training and rapid adaptation to trajectory-specific simulation parameters, such as unknown object material properties, during inference. In addition, we mitigate the problem of error accumulation by directly predicting full simulation trajectories of the mesh nodes instead of iteratively predicting their next-step dynamics. To this end, we represent the simulation using node-level Probabilistic Dynamic Movement Primitives (ProDMPs) (Schaal, 2006; Paraschos et al., 2013; Li et al., 2023), which allows for an efficient encoding of higher-order dynamics at arbitrary temporal resolution. The resulting method, called Movement-Primitive Meta-MeshGraphNet (M3GN), allows efficient generation of context-dependent simulation trajectories that accurately infer and integrate unknown system properties. Figure 1 shows examples for different tasks, while Figure 2 provides an overview of our approach.

To validate the effectiveness of M3GN, we introduce three novel tasks based on challenging deformable object simulations with varying object materials. These include a planar plate bending under stress, and a variety of falling objects that collide with the ground. In addition to these new tasks, we evaluate M3GN on an existing suite of experiments (Linkerhägner et al., 2023).Our results show that our method provides superior simulation accuracy compared to several variants of MeshGraphNet (MGN) Pfaff et al. (2021), the state-of-the-art GNS.[1] Further, the ProDMPs trajectory representation of M3GN reduces the number of required model calls, improving inference runtime by up to 32 times compared to MGN.

In summary, we (i) propose M3GN, a novel GNS that combines meta-learning and movement primitives to predict node-level simulation trajectories based on initial system state contexts; (ii) introduce two challenging deformation prediction tasks involving varying object materials; (iii) evaluate and compare our method to state-of-the-art GNSs, demonstrating superior prediction performance.

---

[1]Code is provided in the supplement. Here, the reviewers can also find a video showing the main results including HD renders of M3GN predictions and comparisons to existing baselines.

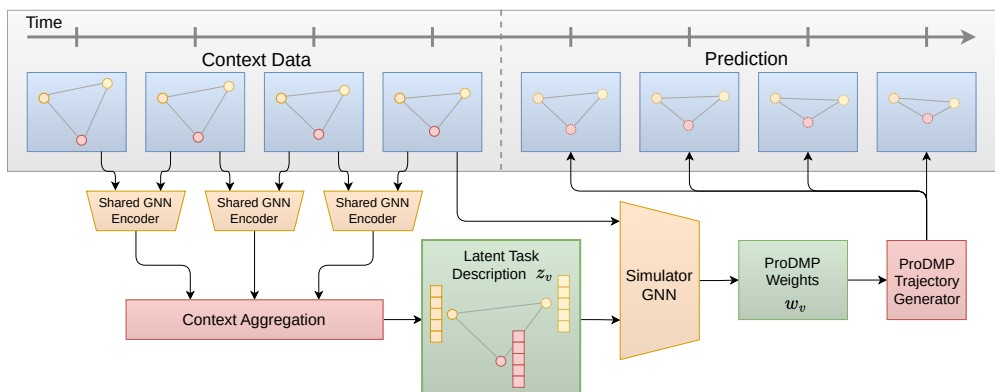

Figure 2: M3GN architecture schematic. Given a context set of initial system states, we calculate node-level latent features for every pair of states using a shared GNN encoder. These feature sets are then aggregated, yielding a node-level latent task description $z_v$. We concatenate this description with the last system state to predict ProDMP weights that are used to compute per-node trajectories.

## 2 RELATED WORK

**Graph Network Simulators.** Deep neural networks for physical simulations can provide significant speedups over traditional simulators while being fully differentiable (Pfaff et al., 2021; Allen et al., 2022a), making them a natural choice for applications like model-based Reinforcement Learning (Mora et al., 2021) and Inverse Design problems (Baqué et al., 2018; Durasov et al., 2021; Allen et al., 2022a). A popular class of learned neural simulators are Graph Network Simulators (GNSs) (Battaglia et al., 2016; Sanchez-Gonzalez et al., 2020). GNSs utilize Message Passing Networks (MPNs), a special type of GNN (Scarselli et al., 2009; Bronstein et al., 2021) that representationally encompasses the function class of many classical solvers (Brandstetter et al., 2022). GNS handle physical data by modeling arbitrary entities and their relations as a graph. Applications of GNSs include particle-based simulations (Li et al., 2019; Sanchez-Gonzalez et al., 2020; Whitney et al., 2023), atomic force prediction (Hu et al., 2021), and fluid dynamic problems (Brandstetter et al., 2022). These models have additionally been applied to the mesh-based prediction of deformable objects (Pfaff et al., 2021; Weng et al., 2021; Han et al., 2022; Fortunato et al., 2022; Linkerhägner et al., 2023). Recent extensions handle rigid objects (Allen et al., 2022b; 2023; Lopez-Guevara et al., 2024) and integrate learned adaptive mesh refinement strategies (Plewa et al., 2005; Freymuth et al., 2023) into the simulator (Wu et al., 2023). Existing work that considers unknown material properties in simulations of deformable objects combines the GNS prediction with point cloud information to improve long-term predictions Linkerhägner et al. (2023). This method requires a constant stream of point clouds to ground the simulation in, but can not aggregate this information into a description of the material properties. Additionally, the DEL method (Wang et al., 2024) integrates physical priors from the Discrete Element Analysis (DEA) framework with learnable graph kernels, addressing the challenges of simulating 3D particle dynamics from 2D images. In the context of larger-scale simulations, foundation models are gaining traction in neural simulation tasks, as exemplified by Aurora (Bodnar et al., 2024), a large-scale model trained on extensive climate data. While Aurora demonstrates impressive performance on atmospheric predictions, including global air pollution and weather forecasts, it requires significantly more data for fine-tuning compared to our approach, which focuses on efficient adaptation with fewer data points. Notably, all previously mentioned GNSs predict system dynamics iteratively from a given state, whereas we directly estimate entire trajectories, improving rollout stability and reducing function calls. Related to our approach is the Equivariant Graph Neural Operator (EGNO) (Xu et al., 2024), which also predicts full trajectories using SE(3) equivariance to model 3D dynamics and capture spatial and temporal correlations. In contrast to the related work, which rely on supervised learning or fine-tuning a foundation model, we employ meta-learning for efficient adaptation to new trajectories.

**Meta-Learning.** Meta-learning (Schmidhuber, 1992; Thrun & Pratt, 1998; Vilalta & Drissi, 2005; Hospedales et al., 2022) extracts inductive biases from a training set of related tasks in order to increase data efficiency on unseen tasks drawn from the same task distribution. In contrast to other

multi-task learning methods, such as transfer learning (Krizhevsky et al., 2012; Golovin et al., 2017; Zhuang et al., 2020), which merely fine-tune or combine standard single-task models, meta-learning makes the multi-task setting explicit in the model architecture (Bengio et al., 1991; Ravi & Larochelle, 2017; Andrychowicz et al., 2016; Volpp et al., 2019; Santoro et al., 2016; Snell et al., 2017). This explicit architecture allows the resulting meta-models to learn *how* to learn new tasks from a small number of example contexts. A popular variant is Model-Agnostic Meta-Learning (MAML) (Finn et al., 2017; Grant et al., 2018; Finn et al., 2018; Kim et al., 2018), which employs standard single-task models and formulates a multi-task optimization procedure. Neural Processes (NPs) (Garnelo et al., 2018a;b; Kim et al., 2019; Gordon et al., 2019; Louizos et al., 2019; Volpp et al., 2021; 2023) instead build on a multi-task model architecture (Heskes, 2000; Bakker & Heskes, 2003) but employ standard gradient based optimization algorithms (Kingma & Ba, 2015; Kingma & Welling, 2014; Rezende et al., 2014; Zaheer et al., 2017). Here, we use Conditional Neural Processes (CNPs) (Garnelo et al., 2018a), which aggregate learned features over a variable-sized context set to yield a latent task description that our downstream GNS is conditioned on. Compared to regular NPs, CNPs assume a deterministic task description, eliminating the need for a distribution over latent variables. This assumption simplifies and accelerates the training process, as our objective is to predict a single precise simulation trajectory from the context set.

## 3 MOVEMENT-PRIMITIVE META-MESHGRAPHNETS

In this section, we present the theoretical foundation of the M3GN method, detailing the algorithmic design choices that guided its development. **Graph Network Simulators.** Consider a graph $\mathcal{G} = (\mathcal{V}, \mathcal{E}, \mathbf{X}_\mathcal{V}, \mathbf{X}_\mathcal{E})$ with nodes $\mathcal{V}$, edges $\mathcal{E}$, and associated vector-valued node and edge features $\mathbf{X}_\mathcal{V}$ and $\mathbf{X}_\mathcal{E}$. An MPN (Sanchez-Gonzalez et al., 2020; Pfaff et al., 2021) consists of $M$ message passing steps, which iteratively update the node and edge features based on the graph topology. Each such step is given as

$$\mathbf{h}_e^{m+1} = f_\mathcal{E}^m(\mathbf{h}_v^m, \mathbf{h}_e^m),$$
$$\mathbf{h}_v^{m+1} = f_\mathcal{V}^m(\mathbf{h}_v^m, \bigoplus_{e \in \mathcal{E}_v} \mathbf{h}_e^{m+1}),$$

where $\mathbf{h}_v^m$ and $\mathbf{h}_e^m$ denote embeddings of the system state per node and edge at message passing iteration $m$, respectively. $\mathcal{E}_v \subset \mathcal{E}$ are the edges connected to $v$. Further, $\bigoplus$ denotes a permutation-invariant aggregation operation such as the sum, the max, or the mean. The functions $f_\mathcal{V}^m$ and $f_\mathcal{E}^m$ are learned Multilayer Perceptrons (MLPs). The network's final output are the node-wise learned representations $\mathbf{h}_v := \mathbf{h}_v^M$ that encode local information of the initial node and edge features.

Conventional GNSs encode the state of the simulated system as a graph, feed it through the MPN, and interpret the per-node outputs as velocities or accelerations (Pfaff et al., 2021). These dynamics are used to forward the simulation in time using, e.g., a forward-Euler integrator (Sanchez-Gonzalez et al., 2020). The graph encodes relative distances and velocities between entities instead of absolute ones, as the resulting equivariance to translation improves generalization (Sanchez-Gonzalez et al., 2020). GNSs usually minimize a next-step Mean Squared Error (MSE) per node during training, adding carefully tuned implicit denoising strategies (Pfaff et al., 2021; Brandstetter et al., 2022) to stabilize long-term predictions. During inference, they compute trajectories by iteratively predicting and integrating their output in an autoregressive fashion. If some simulated objects, like the collider, are known, only the remaining nodes are predicted. Our method instead uses a ProDMP to predict a compact representation of a whole trajectory per system node, reducing the effect of error accumulation, similar to temporal bundling (Brandstetter et al., 2022).

**Probabilistic Dynamic Movement Primitives.** Movement Primitives (MPs) (Schaal, 2006; Paraschos et al., 2013) allow for compact and smooth trajectory representations $y$ via a set of basis functions parameterized by a set of weights $\boldsymbol{w}$. This temporal smoothness is highly beneficial for, e.g., robotic applications (Li et al., 2024; Otto et al., 2022). Recent methods integrate MPs with neural networks to enhance their expressive capabilities (Seker et al., 2019; Bahl et al., 2020; Li et al., 2023). Dynamic Movement Primitives (DMPs) (Schaal, 2006) use a spring-damper dynamical system governed by parameters $\alpha$ and $\beta$. To manipulate the trajectory, an external forcing term $f$ is added, before the system converges to a predefined goal $g$:

$$\tau^2 \ddot{y} = \alpha \left( \beta(g - y) - \tau \dot{y} \right) + f(x), \quad f(x) = x \boldsymbol{\varphi}^\mathsf{T} \boldsymbol{w}. \tag{1}$$

Figure 3: **Left:** M3GN and MGN task setup. Both methods predict mesh positions based on the initial mesh at the anchor time step. M3GN utilizes previous mesh positions and the last step of the collider trajectory for its latent task description, whereas MGN disregards past information and integrates the ground truth collider trajectory into its step-based model. **Right:** Exemplary final simulation steps on the Planar Bending task of M3GN given different context set sizes. A larger context size results in a more accurate **prediction** of the **ground truth** simulation.

Here, $\tau$ influences execution speed, while $f$ depends on the basis functions $\boldsymbol{\varphi}$ in force-space, the weights $\boldsymbol{w}$ and the exponential decaying phase $x$. Solving this equation typically is computationally intensive, particularly when the gradient $dy/d\boldsymbol{w}$ is required (Bahl et al., 2020). ProDMPs (Li et al., 2023) instead solve Equation equation 7 with pre-computed basis functions $\boldsymbol{\Phi}$ in position-space as

$$y(t) = c_1 y_1(t) + c_2 y_2(t) + \boldsymbol{\Phi}(t)^\top \boldsymbol{w}.$$

The term $c_1 y_1(t) + c_2 y_2(t)$ only depends on the initial conditions $[y(t_0), \dot{y}(t_0)]$. ProDMPs thus generate smooth trajectories at an arbitrary temporal resolution from low-dimensional weights $\boldsymbol{w}$. They crucially allow for efficient gradient computation, and can respect different initial conditions such as positions or velocities. We provide an extensive mathematical background of ProDMPs in Appendix A.

**Meta-Learning and Graph Network Simulators.** To enable generalization across tasks with varying properties, we frame GNS as a meta-learning problem. In this setup, each task corresponds to a simulation of a deformable object with unknown material properties. The goal is to learn a simulator that can adapt quickly to a specific scenario using a limited amount of context data. Following the notation of Volpp et al. (2021), the meta-dataset $\mathcal{D} = \mathcal{D}_{1:L}$ consists of simulation trajectories $\mathcal{D}_l = \{\mathcal{G}_{l,1} \ldots \mathcal{G}_{l,T}\}$ of length $T$, where $T$ is the trajectory length. Each simulation step $\mathcal{G}_{l,t} = (\boldsymbol{m}_{l,t}, \boldsymbol{u}_{l,t})$ represents a graph capturing both the deformable object mesh $\boldsymbol{m}_{l,t}$ (describing its position and topology) and an optional rigid collider $\boldsymbol{u}_{l,t}$. Physical proximity is used to define graph edges that model interactions between the deformable object and the collider. At test time, the first $T^c \ll T$ simulation frames, $\mathcal{G}_{l,1:T^c}$, are observed as a context set to predict the remaining trajectory. Following prior work (Pfaff et al., 2021), we assume access to the full collider trajectory during prediction, resulting in the complete *context set*:

$$\mathcal{D}_l^c = \{\mathcal{G}_{l,1}, \ldots \mathcal{G}_{l,T^c}\} \cup \{\boldsymbol{u}_{l,T^c+1}, \ldots, \boldsymbol{u}_{l,T}\}. \tag{2}$$

To provide a clear reference point for discussion, we define the *anchor time step* as the final time step $T^c$ of the context set. The corresponding *anchor graph*, $\mathcal{G}_{l,T^c}$, represents the system's state at this point and serves as the starting state for trajectory prediction by the GNSs. Notably, the anchor graph $\mathcal{G}_{l,T^c}$ alone does not capture the complete system state, as the material properties of the deformable object remain unknown. These properties must be inferred from the prior simulation steps, $\mathcal{G}_{l,1:T^c}$, to enable accurate trajectory predictions. Figure 3 illustrates this setup.

**Model Architecture.** Our model architecture, M3GN, is designed to learn from context data and predict future simulation steps by leveraging a combination of graph network simulation and meta-learning techniques. The architecture consists of two parts: the computation of the latent task description from the context data and the actual graph network simulation of future simulation steps. We base our context processing on the Conditional Neural Process (CNP) (Garnelo et al., 2018a), as it efficiently encodes a latent description over tasks given a set of context observations. Omitting the task index $l$ to avoid clutter, CNPs expect a context set $\{(\boldsymbol{x}_1, \boldsymbol{y}_1), \ldots, (\boldsymbol{x}_{N^c}, \boldsymbol{y}_{N^c})\}$ consisting of inputs $\boldsymbol{x}_i$ and corresponding targets $\boldsymbol{y}_i$. We translate our context set $\mathcal{D}^c$ from Equation 2 to this format by using each graph as an input, and setting its labels as the node-wise velocities. This approach allows the model to focus on dynamics rather than absolute positions, which are more task-specific. Assuming a forward-Euler integration scheme with a time step of 1, we numerically approximate the

velocities as the difference between consecutive simulation steps. The input graph $\boldsymbol{x}_i$ represents the simulation state at time step $i$, including mesh and collider, while $\boldsymbol{y}_i$ encodes the change in positions between consecutive time steps.

$$\boldsymbol{x}_i = \mathcal{G}_i, \qquad \boldsymbol{y}_i = \text{pos}(\mathcal{G}_{i+1}) - \text{pos}(\mathcal{G}_i).$$

To account for the known collider trajectory, we add its relative position as an additional node feature. Specifically, we include the position of the collider at time step $T$, $\text{pos}(\boldsymbol{u}_T)$, relative to its current position, $\text{pos}(\boldsymbol{u}_i)$. Preliminary testing indicated that incorporating the complete future collider trajectory $\text{pos}(\boldsymbol{u}_{T^c}), ...\text{pos}(\boldsymbol{u}_T)$ did not improve the results on our task suite. Given a context set $\mathcal{D}^c$ with anchor time step $T^c$, this results in $T^c - 1$ tuples $(\boldsymbol{x}_i, \boldsymbol{y}_i)$. A shared GNN encoder $h_{\boldsymbol{\theta}}$ computes node-level latent features

$$\boldsymbol{z}_{i,v} = h_{\boldsymbol{\theta}}(\boldsymbol{x}_i, \boldsymbol{y}_i) \in \mathbb{R}^{T^c \times |\mathcal{V}| \times d_z} \tag{3}$$

for each context time step with feature dimension $d_z$. We then aggregate over the context set to obtain $\boldsymbol{z}_v = \bigoplus \boldsymbol{z}_{i,v} \in \mathbb{R}^{|\mathcal{V}| \times d_z}$, using $\bigoplus = \max$ as the aggregation operator. Intuitively, $\boldsymbol{z}_v$ is a representation of the task inferred from the context data $\mathcal{D}^c$, and encodes material properties, future collider movements, and high-level deformations of the simulation. While we use node-level latent features for M3GN, one could additionally aggregate over the nodes to obtain a graph-global task descriptor $\boldsymbol{z} = \bigotimes \boldsymbol{z}_v \in \mathbb{R}^{d_z}$. We explore this choice and different aggregation functions $\bigoplus, \bigotimes$ in Section 4.

Once the task descriptor has been computed, it serves as the input to the predictive stage, enabling simulation of future trajectories. We concatenate the latent description $\boldsymbol{z}_v$ with the node features of the anchor graph $\mathcal{G}_{T^c}$ and subsequently use a GNN $g_{\boldsymbol{\theta}}$ to predict per-node ProDMP weights

$$\boldsymbol{w}_v = g_{\boldsymbol{\theta}}(\mathcal{G}_{T^c}, \boldsymbol{z}_v) \in \mathbb{R}^{|\mathcal{V}| \times d_w}. \tag{4}$$

For certain tasks, incorporating the current node velocities as an additional node feature in the simulator GNN can be advantageous. The ProDMP trajectory generator $f(\boldsymbol{w}_v) \in \mathbb{R}^{T \times |\mathcal{V}| \times d_{\text{world}}}$ transforms the predicted outputs of the simulator GNN into per-node object trajectories over the entire simulation horizon. This approach can be seen as a form of temporal bundling (Brandstetter et al., 2022), requiring a single function call. In comparison, existing GNS train mostly on next-step dynamics and require one call per step during their auto-regressive inference scheme (Pfaff et al., 2021; Allen et al., 2023). The trajectory-level view further allows us to omit noise injection during training, which MGN requires to generalize from learned next-step predictions to multi-step rollouts during inference. We provide a visualization of our model architecture in Figure 2 and refer to Appendix B for further details.

**Meta Training.** The goal of meta-learning is to automatically encode inductive biases towards the task distribution extracted from the meta-dataset $\mathcal{D}$ into the task-global parameter $\boldsymbol{\theta}$. To this end, we minimize the negative conditional log probability (Garnelo et al., 2018a)

$$\mathcal{L}(\boldsymbol{\theta}) = -\mathbb{E}_{l \sim 1:L} \Big[ \mathbb{E}_{T^c \sim T_{\min}:T_{\max}} \big[ \log p_{\boldsymbol{\theta}}(\text{pos}(\boldsymbol{m}_{l,1:T}) \mid \mathcal{D}_l^c) \big] \Big]. \tag{5}$$

Each training batch consists of a task $\mathcal{D}_l$ for which we sample a context size $T^c$ uniformly between $T_{\min}$ and $T_{\max}$ to ensure that the model learns to handle different context set sizes. We then compute the latent task descriptor $\boldsymbol{z}_v$ and subsequently the predicted node trajectories $f(g_{\boldsymbol{\theta}}(\mathcal{G}_{l,T^c}, \boldsymbol{z}_v))$ as described in Equation 3 and Equation 4. The likelihood $p_{\boldsymbol{\theta}}$ is defined to be the Gaussian

$$p_{\boldsymbol{\theta}}(\text{pos}(\boldsymbol{m}_{l,1:T}) \mid \mathcal{D}_l^c) = \mathcal{N}(\text{pos}(\boldsymbol{m}_{l,1:T}) \mid f(g_{\boldsymbol{\theta}}(\mathcal{G}_{l,T^c}, \boldsymbol{z}_v)), \boldsymbol{\sigma}_o). \tag{6}$$

Since the training simulations are not affected by noise, we are not modeling the output variance and set it to $\boldsymbol{\sigma}_0 = 1$. Together with taking the mean over the nodes and time steps to stabilize training, optimizing the Gaussian log likelihood from Equation 6 is equivalent to minimizing the MSE

$$\log p_{\boldsymbol{\theta}}(\text{pos}(\boldsymbol{m}_{l,1:T}) \mid \mathcal{D}_l^c) \simeq \frac{1}{T\,|\mathcal{V}|\,d_{\text{world}}} \sum_{t,v,i} \Big( \text{pos}(\boldsymbol{m}_{l,t})_{v,i} - f(g_{\boldsymbol{\theta}}(\mathcal{G}_{l,T^c}, \boldsymbol{z}_v))_{t,v,i} \Big)^2.$$

The whole architecture is trained end-to-end using the loss $\mathcal{L}(\boldsymbol{\theta})$ from Equation 5. After the meta-training, we fix $\boldsymbol{\theta}$, which now encodes inductive biases towards the meta-data $\mathcal{D}$.

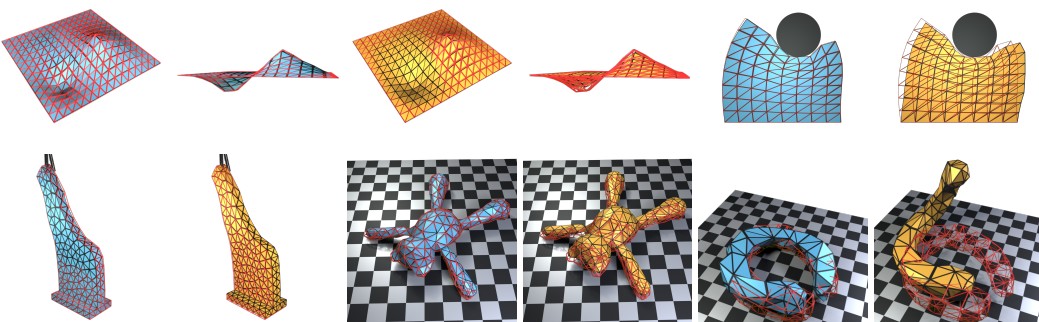

Figure 4: Comparison of the final simulation step between **M3GN** and **MGN** on all datasets. From (**Left**) to (**Right**): (**Top**) *Planar Bending* and *Deformable Plate* with an anchor time step of 2. (**Bottom**) *Tissue Manipulation* with a context size of 6, and *Falling Teddy Bear* and *Mixed Objects Falling* with an anchor time step of 20. M3GN provides much better alignment to the **ground truth** simulation on all tasks, except for *Tissue Manipulation*, where MGN also solves the task well.

## 4 EXPERIMENTS

**Setup.** Our experimental setup largely follows previous work (Linkerhägner et al., 2023). The graph representation views the mesh vertices as nodes and adds edges according to the mesh topology within an object, and based on Euclidean distance between different objects. We employ one-hot encoding to differentiate between deformable objects and colliders, omitting explicit world edges (Pfaff et al., 2021). The edges additionally contain the relative distances between their nodes.

For both the context MPN as well as the simulator MPN, we use 15 message passing steps. Each message passing step uses separate 1-layer MLPs with a latent dimension of 128 and LeakyReLU activations for its node and edge updates. We evaluate the *Full Rollout MSE*, which is calculated as the average of all simulation MSEs following the anchor time step, and the 10-*step MSE*, which is the average performance for the next 10 simulation steps following the anchor time step. Both metrics are averaged over all test set trajectories. We report the interquartile mean and bootstrapped confidence intervals (Agarwal et al., 2021) over 8 random seeds for each experiment. We evaluate the metrics for various context sizes ranging from 2 to 30 steps, always setting the anchor time step to the last context mesh position. Appendix C provides additional details on our experimental setup.

**Datasets.** We validate our method on five different simulation datasets based on three different mesh-based physics simulators. All task spaces are normalized to $[-1, 1]^3$. These include a 2D *Deformable Plate (DP)* task and a 3D *Tissue Manipulation (TM)* task (Linkerhägner et al., 2023) In both datasets, the material property, Poisson's ratio (Lim, 2015) was randomized. *Deformable Plate* simulates different trapezoids that are deformed by a circular collider with constant velocity and varying size and starting position. Each trajectory consists of a mesh with 81 nodes that is deformed over 39 time steps. *Tissue Manipulation* considers a surgical robotics scenario where a piece of tissue is deformed by a gripper. The gripper is attached to a fixed object position and moves in a random direction with constant velocity. The mesh comprises 361 nodes and the simulation has 100 steps.

We further propose three additional 3D datasets. *Planar Bending (PB)* simulates the bending of a 2D plane when two constant forces perpendicular to the plane are applied at different positions. As the Young's modulus is varied between sheets, this dataset constitutes a simplified stamp forming process, as common in material engineering (Zimmerling et al., 2022). The simulations are generated with Abaqus (Smith, 2009), comprising 50 time steps and a plate with 225 nodes. We test two different data splits: The in-distribution (ID) split uses material properties in the test set that the model has seen during training, while the out-of-distribution (OOD) split uses Young's modulus values outside the training domain for the test dataset.

The other two tasks place a randomly rotated deformable object at a specific height and let it fall to and collide with the ground. *Falling Teddy Bear (FTB)* considers the titular teddy bear as its only object, whereas six different objects are considered for *Mixed Objects Falling (MOF)*. Each trajectory assigns a random Poisson's ratio and random Young's modulus to the falling object, thus influencing its deformation upon contact with the floor. Each trajectory consists of 200 time steps. The object

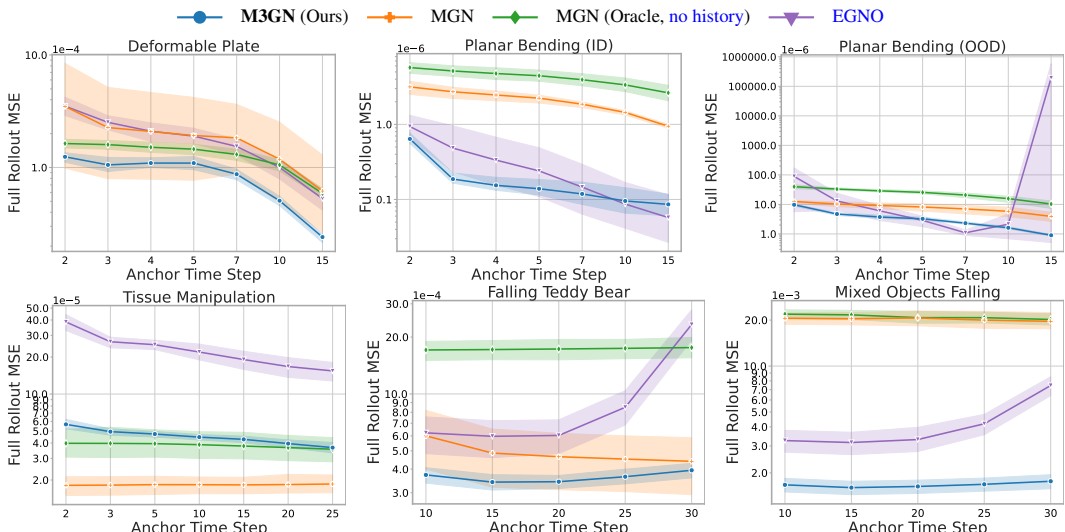

Figure 5: Log-scale MSE over full rollouts for different methods under all tasks, including an additional figure for Planar Bending task using an out-of-distribution (OOD) test dataset of material properties. Overall, M3GN steadily improves its performance when provided with additional context information and a later anchor time step. Our method generally improves over MGN, likely due to the MP trajectory representation, and outperforms MGN (Oracle) when provided with sufficient context information. Equivariant Graph Neural Operator (EGNO) generally performs unstable for later anchor time steps and can only compete in the *Planar Bending (ID)* task.

meshes have up to 350 nodes and are shown in Figure 9 in the appendix. For simplicity, we only consider the triangular surface meshes for the experimental setup. Further information about the graph encoding are given in Appendix B.1, while we provide detailed information of the dataset sizes and preprocessing steps in Appendix D.

**Baselines and Ablations.** We compare our method to MGN(Pfaff et al., 2021), evaluating its performance both with and without additional material property information provided as a node feature. When this *oracle* information is available, the simulation becomes deterministic with respect to the initial system state. Importantly, we never supply this node feature to M3GN.

MGN generates the next mesh state by iteratively predicting the velocities for the current simulation step. It is trained to minimize the 1-step MSE over node velocities, incorporating Gaussian input noise during training (Brandstetter et al., 2022). This noise serves to mitigate error accumulation and stabilize auto-regressive rollouts during inference. To ensure a fair comparison, we adopt the same hyperparameters as our method and fine-tune the input noise level for each task, maximizing MGN's performance.

We also explore the effect of incorporating historical information, specifically previous velocities, as node features for both MGN and M3GN. For MGN, using both the current and previous velocities improves performance significantly on many tasks. For M3GN, including only the current velocity yields similar benefits. The results of a hyperparameter optimization on the validation split are presented in Figure 10 in the Appendix. Additionally, Table 1 summarizes the specific history configurations used for each method and task, along with other relevant hyperparameters.

As an additional baseline, we compare our approach to the Equivariant Graph Neural Operator (EGNO) method, which employs equivariant message-passing layers and predicts the remaining simulation steps in a single pass, closely aligning with our setup. However, training EGNO proved unstable with 15 message-passing steps, and the best results were achieved using only 5 steps. We hypothesize that this instability may stem from the longer prediction horizon of up to 200 steps in our experiments, as the baseline was originally evaluated on tasks with a much shorter prediction horizon of only 8 steps. Further details on the implementation of these baselines can be found in Appendix C.

Additionally, we ablate different design choices of M3GN on *Planar Bending* and *Deformable Plate*. To investigate the effect of the meta-learning approach, we train an MGN (MP) variant that uses

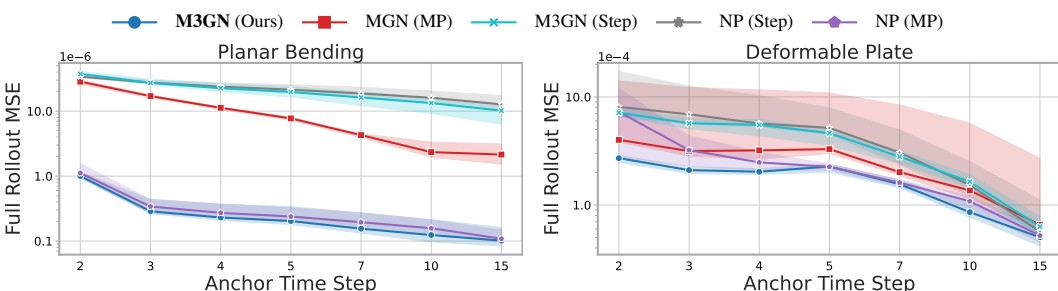

Figure 6: Log-scale MSE over full rollouts for the *Planar Bending* (**Left**) and *Deformable Plate* (**Right**) tasks for different meta-learning and MP variants. Using a ProDMP representation for MGN improves performance. CNPs and NPs with a next-step prediction do not improve over standard MGN. A NP instead of an CNP architecture for M3GN slightly reduces performance.

ProDMPs predictions, but omits a context aggregation and thus has no latent task description $z_v$. Similarly, we compare to M3GN (Step-based), which performs a next-step prediction of the dynamics instead of predicting ProDMP parameters, but otherwise follows the CNP training scheme to learn a latent task description. Finally, we compare the deterministic CNP approach to both MP and step-based probabilistic Neural Process (NP) approaches. Here, we get diagonal Gaussian distributions as the outputs of the context MPN, which we aggregate using Bayesian context aggregation (Volpp et al., 2021). We further investigate if node aggregation of the latent task description is beneficial by applying a maximum aggregation of the node features before the context aggregation. While standard CNPs require a permutation-invariant context aggregation, our context has a temporal structure. We thus experiment with a small transformer model with 4 transformer blocks, 4 attention heads, temporal encoding and a latent dimension of 32 as an aggregator. The transformer takes the sequence of outputs of the context MPN and predicts the aggregated node-level task description $z_v$.

**Results.** Figure 4 visualizes exemplary final simulation steps for M3GN and MGN for all tasks. M3GN aggregates context information that it uses to condition node-level ProDMP representations of the simulated trajectory. This approach leads to accurate simulations, providing much better alignment to the ground truth simulation than the step-based MGN on all tasks. Appendix E.3 shows visualizations of full simulation rollouts for all tasks and methods.[2]

Figure 5 shows the full rollout MSE for M3GN, MGN and MGN (Oracle) on all tasks. In general, the additional material information improves performance of MGN (Oracle) on both datasets compared to MGN. A later anchor time step improves performance for all methods, presumably because the remaining simulation is shorter. On *Deformable Plate*, M3GN surpasses MGN (Oracle) for a context size of 10, likely because the additional context improves the latent task description. For the *Planar Bending* task, M3GN significantly outperforms the step-based baselines across context sizes, likely because the temporal smoothness of the ProDMP trajectory is a strong inductive bias for the gradual bending of the simulated plane. Furthermore, M3GN generalizes well to the OOD task, while the MGN methods fail to extrapolate to unseen material properties.

Next, in a more difficult task, such as *Tissue Manipulation*, increasing the context size greatly improves performance for M3GN, whereas the step-based methods only slightly benefit from a later anchor time step. On the last two tasks, i.e. *Falling Teddy Bear* and *Mixed Objects Falling*, the step-based MGN methods fail to provide accurate long-term simulations. Here, the predicted trajectories usually qualitatively deviate from the ground truth, either causing an object drift or a misalignment of, e.g., teddy limbs, to the point that providing additional material information only yields marginal improvements. For M3GN, the ProDMP's temporally consistent movements combined with context aggregation to provide consistent simulations alleviates these issues, significantly improving over the baselines. The bottom of Figure 4 provides examples. Interestingly, providing more context information does not improve performance for either of these two tasks. A likely reason for this behavior is that most early context steps consist of a falling object, resulting in similar graph representations, which may cause the GNN context encoders to overfit.

---

[2]Videos of these simulations are in the supplementary material.

Figure 6 provides evaluations for various ablations, further supporting these results. Equipping MGN with a ProDMP representation improves performance, especially for *Planar Bending*. Combining NPs and ProDMPs uniquely leads to the strong performance of M3GN, suggesting that the latent task description extracted from the context set is particularly well-suited for trajectory-level representations of the simulations. Performance decreases slightly when using an NP instead of a CNP, indicating that the latent distribution in NPs is not beneficial for our GNS setup. Additional ablations in Figure 11 of the Appendix show similar performance for different aggregation schemes. Our node-level maximum context aggregation is the simplest and works slightly better than the alternatives for *Planar Bending*.

To provide a more detailed understanding of our model's performance, we include additional plots in Appendix E showing the Mean Squared Error (MSE) over time for the trajectory, rather than just the final averaged MSE. These plots demonstrate the temporal progression of errors, offering insights into the model's behavior throughout the simulation. We additionally report the 10-step MSE in Appendix E.2, finding that the relative improvement of M3GN compared to the baselines matches or exceeds that on the full rollout MSE for all tasks. Furthermore, we present a visualization of the latent space for M3GN in the Appendix, Figure 20, which reveals that simulations with similar material properties are clustered together. This structure in the latent space highlights the model's ability to effectively differentiate between different material behaviors while preserving the relationships between similar properties. Finally, we compare the inference speed between M3GN and MGN in Figure 7. The ProDMPs trajectory representation of M3GN decreases the amount of required model calls, resulting in an inference-time speedup of up to 32 times compared to MGN, and up to 400 times compared to the ground truth simulators (Gth Simulators). We encode the context set for M3GN in parallel, resulting in a relatively minor cost for the context computation and aggregation for all tasks. In addition, our M3GN requires only one GNN forward pass via ProDMP to compute the full trajectory, enabling much faster inference. While MGN does not perform any context processing, its iterative rollout is inherently sequential and requires multiple forward passes.

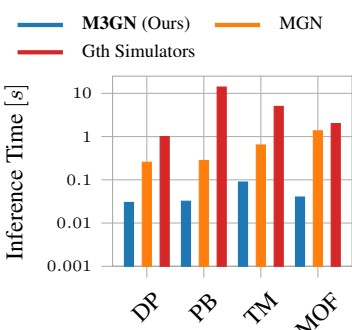

Figure 7: Runtime comparison on four tasks between the learned methods and the different ground truth simulators. Note the log scale on the y-axis.

## 5 CONCLUSION

We introduce Movement-Primitive Meta-MeshGraphNet (M3GN), a novel Graph Network Simulator that combines movement primitives and trajectory-level meta-learning for efficient and accurate long-term predictions in physical simulations. Our method dynamically adapts to provided context information during inference, allowing for an accurate prediction of deformations under unknown object properties. Additionally, it effectively addresses the issue of error accumulation while reducing the number of required simulator function calls. To validate the effectiveness of M3GN, we propose three novel deformation prediction tasks with uncertain material properties. Results on these tasks and existing datasets show that our method consistently outperforms a strong Graph Network Simulators baseline, even when providing the baseline with oracle information about the material property.

**Limitations and Future Work.** We currently consider each trajectory as a task, and require initial states of this trajectory as a context set during inference. As generating such data is often impractical, we plan to group simulations with the same system properties into the same task. This adaptation will enable data-efficient generalization to unseen properties regardless of the underlying simulated objects. We also plan to integrate online re-planning of trajectories, predicting trajectory segments with every model forward pass. This process may increase coordination between simulated nodes across segments, while maintaining the benefits of a compact multi-step trajectory representation.

**Broader Impact Statement** Our proposed Graph Network Simulator can positively impact various fields relying on computational modeling and simulation by significantly reducing computational cost compared to traditional simulators while providing accurate simulations. However, efficient and accurate simulation of physical systems also comes with potential negative impacts, such as, e.g., the development of advanced weapon models.

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

## A MATHEMATICAL FORMULATIONS OF MOVEMENT PRIMITIVES

We provide an overview of the probabilistic dynamic movement primitives (ProDMP) formulations utilized in this paper, starting with the foundational methods: Dynamic Movement Primitives (DMPs) and Probabilistic Movement Primitives (ProMPs).

### A.1 DMPs

Schaal (2006) introduced Dynamic Movement Primitives (DMPs), which integrate a forcing term into a dynamical system to generate smooth trajectories from given initial conditions[3], such as a robot's position and velocity at a particular time. A DMP trajectory is governed by a second-order linear ordinary differential equation (ODE) as follows:

$$\tau^2 \ddot{y} = \alpha(\beta(g - y) - \tau \dot{y}) + f(x), \quad f(x) = x \frac{\sum \varphi_i(x) w_i}{\sum \varphi_i(x)} = x \boldsymbol{\varphi}_x^\mathsf{T} \boldsymbol{w}, \tag{7}$$

where $y = y(t)$, $\dot{y} = \mathrm{d}y/\mathrm{d}t$, and $\ddot{y} = \mathrm{d}^2 y/\mathrm{d}t^2$ denote the position, velocity, and acceleration of the system at a specific time $t$, respectively. Constants $\alpha$ and $\beta$ are spring-damper parameters, $g$ is the goal attractor, and $\tau$ is a time constant modulating the speed of trajectory execution.

The functions $\varphi_i(x)$ represent the basis functions for the forcing term, as shown in Fig. 8a, while the phase variable $x = x(t) \in [0, 1]$ captures the execution progress. The trajectory's shape is determined by the weight parameters $w_i \in \boldsymbol{w}$ for $i = 1, \ldots, N$ and the goal term $g$. The trajectory $[y_t]_{t=0:T}$ is typically computed by numerically integrating the dynamical system from the start to the endpoint. However, this numerical process is computationally expensive (Bahl et al., 2020; Li et al., 2023), as its cost scales with the trajectory length and the resolution of the numerical integration.

### A.2 PROMPS

Paraschos et al. (2013) introduced the Probabilistic Movement Primitives (ProMPs) framework for modeling trajectory distributions, effectively capturing both temporal and inter-dimensional correlations. Unlike DMPs, which rely on a forcing term, ProMPs directly model the desired trajectory and its distribution using a linear basis function representation. Given a weight vector $\boldsymbol{w}$ or a weight vector distribution $p(\boldsymbol{w}) \sim \mathcal{N}(\boldsymbol{w}|\boldsymbol{\mu_w}, \boldsymbol{\Sigma_w})$, the corresponding trajectory or trajectory distribution is computed as follows:

$$\text{Compute Trajectory:} \quad [y_t]_{t=0:T} = \boldsymbol{\Phi}^\mathsf{T} \boldsymbol{w}, \tag{8}$$

$$\text{Compute Distribution:} \quad p([y_t]_{t=0:T}; \; \boldsymbol{\mu_y}, \boldsymbol{\Sigma_y}) = \mathcal{N}(\boldsymbol{\Phi}^\mathsf{T} \boldsymbol{\mu_w}, \; \boldsymbol{\Phi}^\mathsf{T} \boldsymbol{\Sigma_w} \boldsymbol{\Phi}). \tag{9}$$

Here, the matrix $\boldsymbol{\Phi}$ contains the basis functions for each time step $t \in [0, T]$, shown in Fig. 8a. The trajectory shape is determined by the weight parameters $w_i \in \boldsymbol{w}$ through matrix-vector multiplication. Despite their simplicity and computational efficiency, ProMPs lack an intrinsic dynamic system, limiting their ability to specify a given initial condition for a trajectory or predict smooth transitions between two ProMP trajectories with differing parameter vectors.

### A.3 PRODMPS

**Solving the ODE underlying DMPs** Li et al. (2023) observed that the governing equation of DMPs, as described in Eq. (7), admits an analytical solution. We re-express the original ODE from Eq. (7) and its homogeneous counterpart in standard ODE forms as follows:

$$\text{Non-homo. ODE:} \quad \ddot{y} + \frac{\alpha}{\tau} \dot{y} + \frac{\alpha\beta}{\tau^2} y = \frac{f(x)}{\tau^2} + \frac{\alpha\beta}{\tau^2} g \equiv F(x, g), \tag{10}$$

$$\text{Homo. ODE:} \quad \ddot{y} + \frac{\alpha}{\tau} \dot{y} + \frac{\alpha\beta}{\tau^2} y = 0. \tag{11}$$

---

[3]In mathematics, an initial condition refers to the value of a function or its derivatives at a starting point, which can be specified at any time, not necessarily at $t = 0$.

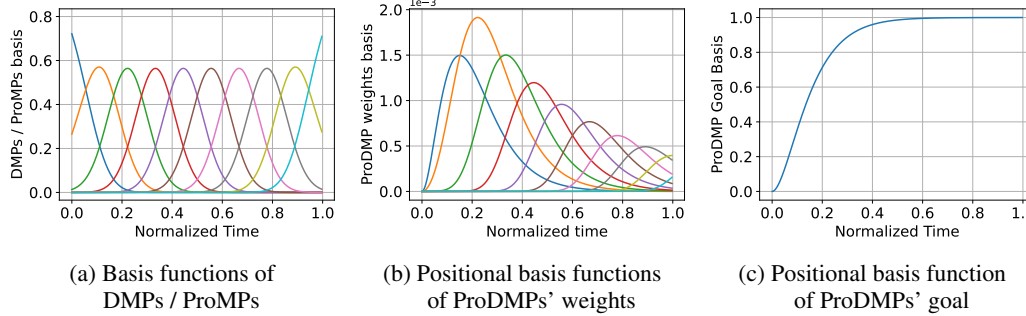

(a) Basis functions of
DMPs / ProMPs

(b) Positional basis functions
of ProDMPs' weights

(c) Positional basis function
of ProDMPs' goal

Figure 8: Illustration of basis functions used in MP methods. (a) Normalized radial basis functions used in DMPs in Eq.(7) and ProMPs in Eq.(8), respectively. (b) Positional basis functions of ProDMPs' weights $\boldsymbol{w}$ and (c) ProDMPs' goal $g$ in Eq.(17). In ProDMPs, $g$ is concatenated with the weights vector $\boldsymbol{w}$ and treated as one dimension of the resulting vector $\boldsymbol{w}_g$. Both weights and goal basis functions are computed from solving the DMPs' underlying ODE, following the procedure from Eq.(12) to Eq.(16)

The solution to this ODE is essentially the position trajectory, and its time derivative yields the velocity trajectory. They are formulated through several time-dependent function as:

$$y = \begin{bmatrix} y_2 \boldsymbol{p_2} - y_1 \boldsymbol{p_1} & y_2 q_2 - y_1 q_1 \end{bmatrix} \begin{bmatrix} \boldsymbol{w} \\ g \end{bmatrix} + c_1 y_1 + c_2 y_2 \tag{12}$$

$$\dot{y} = \begin{bmatrix} \dot{y}_2 \boldsymbol{p_2} - \dot{y}_1 \boldsymbol{p_1} & \dot{y}_2 q_2 - \dot{y}_1 q_1 \end{bmatrix} \begin{bmatrix} \boldsymbol{w} \\ g \end{bmatrix} + c_1 \dot{y}_1 + c_2 \dot{y}_2. \tag{13}$$

Here, the learnable parameters $[\boldsymbol{w}, g]^T$ which control the shape of the trajectory, are separable from the remaining time-dependent functions $y_1, y_2, \boldsymbol{p}_1, \boldsymbol{p}_2, q_1, q_2$. These functions are computed by solving the ODE in Eq. (10), (11):

$$y_1(t) = \exp\left(-\frac{\alpha}{2\tau} t\right), \qquad\qquad y_2(t) = t \exp\left(-\frac{\alpha}{2\tau} t\right), \tag{14}$$

$$\boldsymbol{p}_1(t) = \frac{1}{\tau^2} \int_0^t t' \exp\left(\frac{\alpha}{2\tau} t'\right) x(t') \boldsymbol{\varphi}_x^\intercal \mathrm{d}t', \qquad \boldsymbol{p}_2(t) = \frac{1}{\tau^2} \int_0^t \exp\left(\frac{\alpha}{2\tau} t'\right) x(t') \boldsymbol{\varphi}_x^\intercal \mathrm{d}t', \tag{15}$$

$$q_1(t) = \left(\frac{\alpha}{2\tau} t - 1\right) \exp\left(\frac{\alpha}{2\tau} t\right) + 1, \qquad q_2(t) = \frac{\alpha}{2\tau} \left[\exp\left(\frac{\alpha}{2\tau} t\right) - 1\right]. \tag{16}$$

Here, the function $y_1, y_2$ are the complementary solutions to the homogeneous ODE presented in Eq.(11), with $\dot{y}_1, \dot{y}_2$ their time derivatives respectively.

It's worth noting that $\boldsymbol{p}_1$ and $\boldsymbol{p}_2$ cannot be derived analytically due to the complexity of the forcing basis terms $\boldsymbol{\varphi}_x$. Consequently, these terms must be computed numerically. However, isolating the learnable parameters, namely $\boldsymbol{w}$ and $g$, enables the reuse of other time-dependent functions across all generated trajectories.

ProDMPs identify these reusable terms as the position and velocity basis functions, denoted by $\boldsymbol{\Phi}(t)$ and $\dot{\boldsymbol{\Phi}}(t)$, respectively. Fig. 8b and Fig. 8c illustrate the resulting position basis functions for the weights $\boldsymbol{w}$ and the goal $g$, respectively. These functions are pre-computed offline and treated as constants during online learning. When $\boldsymbol{w}$ and $g$ are combined into a concatenated vector, represented as $\boldsymbol{w}_g$, the position and velocity trajectories can be expressed in a manner similar to that used by ProMPs:

$$\textbf{Position:} \quad y(t) = \boldsymbol{\Phi}(t)^\intercal \boldsymbol{w}_g + c_1 y_1(t) + c_2 y_2(t), \tag{17}$$

$$\textbf{Velocity:} \quad \dot{y}(t) = \dot{\boldsymbol{\Phi}}(t)^\intercal \boldsymbol{w}_g + c_1 \dot{y}_1(t) + c_2 \dot{y}_2(t). \tag{18}$$

In the main paper, for simplicity and notation convenience, we use $\boldsymbol{w}$ instead of $\boldsymbol{w}_g$ to describe the parameters and goal of ProDMPs.

**Trajectory's Initial Condition** The coefficients $c_1$ and $c_2$ are solutions to the initial value problem defined by Eqs.(17)(18). Assuming the trajectory starts at time $t_b$ with position $y_b$ and velocity $\dot{y}_b$, we denote the values of the complementary functions and their derivatives at the condition time $t_b$ as $y_{1_b}, y_{2_b}, \dot{y}_{1_b}$ and $\dot{y}_{2_b}$. Similarly, the values of the position and velocity basis functions at $t_b$ are denoted as $\boldsymbol{\Phi}_b$ and $\dot{\boldsymbol{\Phi}}_b$ respectively. Using these notations, $c_1$ and $c_2$ are computed as:

$$\begin{bmatrix} c_1 \\ c_2 \end{bmatrix} = \begin{bmatrix} \frac{\dot{y}_{2_b} y_b - y_{2_b} \dot{y}_b}{y_{1_b} \dot{y}_{2_b} - y_{2_b} \dot{y}_{1_b}} + \frac{y_{2_b} \dot{\boldsymbol{\Phi}}_b^{\mathsf{T}} - \dot{y}_{2_b} \boldsymbol{\Phi}_b^{\mathsf{T}}}{y_{1_b} \dot{y}_{2_b} - y_{2_b} \dot{y}_{1_b}} \boldsymbol{w}_g \\ \frac{y_{1_b} \dot{y}_b - \dot{y}_{1_b} y_b}{y_{1_b} \dot{y}_{2_b} - y_{2_b} \dot{y}_{1_b}} + \frac{\dot{y}_{1_b} \boldsymbol{\Phi}_b^{\mathsf{T}} - y_{1_b} \dot{\boldsymbol{\Phi}}_b^{\mathsf{T}}}{y_{1_b} \dot{y}_{2_b} - y_{2_b} \dot{y}_{1_b}} \boldsymbol{w}_g \end{bmatrix}. \tag{19}$$

**Set Goal Convergence Relative to Initial Condition** The goal attractor $g$ in the ProDMPs framework represents an asymptotic convergence point for the dynamical system as $t \to \infty$, typically defined as an absolute coordinate. However, the goal term can also be modeled relative to the initial position $y_b$. In this approach, the relative goal $g_{\text{rel}}$ is predicted, and its absolute counterpart is computed as $g_{\text{abs}} = g_{\text{rel}} + y_b$. This approach is particularly useful for predicting the goal in the coordinate system relative to a node's starting position. Since we aim to achieve a translation-equivariant approach (where absolute node positions are encoded as relative edge features between nodes), predicting relative goal positions aligns well with this design principle.

## B  ARCHITECTURE AND METHOD DETAILS

This section offers detailed insights into our methodology and the architectural decisions guiding our approach.

### B.1  GRAPH ENCODINGS

In processing the initial graph $\mathcal{G}_{*,T^c}$, we create edges between the mesh and the collider based on a radius graph. Specifically, we connect mesh and collider nodes for *Deformable Plate* and *Tissue Manipulation* if their euclidean distance is smaller than $0.3$. In the *Tissue Manipulation* task, the collider is given as a single node which is connected to the tip of the tissue. It marks the grasping point of a gripper. In the *Planar Bending* task, we add an additional node feature to the nodes which get directly influenced by the external force. Therefore, no collider is used in this task. For the *Falling Teddy Bear* and the *Mixed Objects Falling* task, we implicitly model the ground as a collider by adding the current $z$ position of every node to its node features (Sanchez-Gonzalez et al., 2018). This quantity gets updated for the step-based methods.

### B.2  PRODMP DETAILS

Initialization of ProDMPs necessitates node velocities for the anchor time step $T^c$. We employ a linear approximation, leveraging data from the previous time step $T^c - 1$.

Similar to the relative encoding of node positions in the MPN, we employ a technique in ProDMP to derive relative trajectories. Initially, we integrate a relative goal position as part of the node weights $\boldsymbol{w}_v$. Utilizing this approach, trajectories commence from the origin and traverse towards their respective relative goals. Subsequently, we adjust all positions by the initial position. This strategy fosters model generalization across various nodes.

The parameter $\tau$, as described in Equation 7, is learned globally across all tasks using a compact MLP. The model's final layer employs a scaled sigmoid function for parameter estimation.

## C  EXPERIMENTAL PROTOCOL

In order to promote reproducibility, we provide details of our experimental methodology. Table 1 presents the hyperparameters used in our experiments. For a comprehensive description of the creation of all datasets, please refer to Appendix D.

The training took place on an NVIDIA A100 GPU, with each method given the same computation budget of 48 hours, except for the *Planar Bending* task, where the computation budget was set to 24

hours. Consequently, the number of epochs varied, as the batching differed significantly between the trajectory-based method M3GN and the step-based MGN. We adapted the batchsize of the step-based methods in order to use the GPU memory efficiently. M3GN is always trained on one full trajectory per batch. Here, the whole context is processed in parallel and the remaining trajectory is predicted and compared to the ground truth.

We conducted a multi-staged grid-based hyperparameter search for the learning rate, input noise, and other hyperparameters as the latent task description dimension. In general, we optimized all methods on all tasks separately, however, we noticed that over different tasks and methods some parameters had the same best configuration. We did not use the test data for this, but tuned all hyperparameters on a separate validation split. This split was also used to determine the best epoch checkpoint to mitigate any overfitting effects.

In the end, all methods worked well with a learning rate of $5.0 \times 10^{-4}$ except in the *Planar Bending* task. Here, our hyperparameter optimization indicated that the trajectory based methods benefit from a smaller learning rate of $1.0 \times 10^{-5}$. For MGN, we experimented with different input noise scales. Notably, for the *Deformable Plate* and the *Planar Bending* task, a smaller noise scale improved performance significantly. In the falling objects tasks, we also explored second-order predictions, such as node accelerations, instead of velocity predictions. Following the approach in Pfaff et al. (2021), we adjusted the labels accordingly and conducted preliminary evaluations. However, since direct velocity predictions yielded superior results, we opted for them as our final approach, as presented in the main paper.

### C.1   EGNO TRAINING

For the Equivariant Graph Neural Operator (EGNO) method, we used the original code from Xu et al. (2024) for the model implementation. Since EGNO can only predict for a fixed next horizon, we cut the remaining prediction when using a later anchor time step. This is done during training and evaluation.

### C.2   MGN TRAINING

We mainly follow Pfaff et al. (2021) for the training of the MGN baseline. The only difference is the incorporation of current and historic velocity node features. Pfaff et al. (2021) consider this in their experiments but they show in their experiment suite that it does not improve the results and can lead to overfitting. This is different to our results. For us, on all tasks except *Mixed Objects Falling*, adding the current and historic velocities of nodes improves the results. We follow the Gaussian random walk noise injection for the velocity features from Sanchez-Gonzalez et al. (2020).

## D   DATASETS AND PREPROCESSING INFORMATION

In this section, we give detailed information about the datasets we used. We report a general overview of all datasets in Table 2. Here each dataset is abbreviated for brevity as the following:

Table 2 lists in detail the datasets used in the paper. Each dataset is abbreviated for brevity and explained as follows:

- **PB.**: *Planar Bending*
- **DP.**: *Deformable Plate*
- **TM.**: *Tissue Manipulation*
- **FTB.**: *Falling Teddy Bear*
- **MOF.**: *Mixed Objects Falling*

### D.1   PLANAR BENDING

We select 9 different Young's modulus ranging between 10 and 1000 from a very deformable to an almost stiff sheet. Then, per material, we compute 100 simulations using Abaqus where the positions

Table 1: Table listing the hyperparameters and configurations of the experiments

| Parameter | Value |
|---|---|
| Node feature dimension | 128 |
| Latent task description dimension | 64 |
| Decoder hidden dimension | 128 |
| Message passing blocks | 15 |
| Message passing blocks (EGNO) | 5 |
| GNN Aggregation function | Mean |
| GNN Activation function | Leaky ReLU |
| M3GN Context Aggregation method | Max Aggr. |
| M3GN Latent Node Aggregation method | No Aggr. |
| Learning rate | $5.0 \times 10^{-4}$ |
| Learning rate (*Plan. Bend.* MP-based methods) | $1.0 \times 10^{-5}$ |
| Number of ProDMP basis functions | 30 |
| ProDMP $\tau$ | learned range: $[0.3, 3.0]$ |
| ProDMP Relative start position | True |
| MGN input mesh noise | 0.01 |
| MGN input mesh noise (*Def. Plate*) | 0.001 |
| MGN input mesh noise (*Plan. Bend.*) | 0.0001 |
| MGN history length (all tasks except *Mixed Objects Fall* | 2 |
| MGN history length (*Mixed Objects Fall*) | 0 |
| M3GN history length (all tasks except *Tiss. Man.* and *Plan. Bend. (OOD)*) | 1 |
| M3GN history length (*Tiss. Man.* and *Plan. Bend. (OOD)*) | 0 |
| Minimum/Maximum Train Context Size (*Plan. Bend.*) | 2 / 15 |
| Minimum/Maximum Train Context Size (*Deformable Plate*) | 2 / 15 |
| Minimum/Maximum Train Context Size (*Tissue Manipulation*) | 2 / 40 |
| Minimum/Maximum Train Context Size (*Falling Teddy Bear*) | 10 / 50 |
| Minimum/Maximum Train Context Size (*Mixed Objects Fall*) | 10 / 50 |
| Threshold to create collider-mesh edge | 0.3 |

Table 2: Table listing the datasets and their configurations

| Name | Train/Val/Test Splits | Number of steps | Number of Nodes | Collider interaction |
|---|---|---|---|---|
| PB. | 630/135/135 | 50 | 225 | External Force |
| DP. | 675/135/135 | 39 | 81 | Rigid Collider |
| TM. | 600/120/120 | 100 | 361 | Grasping point |
| FTB. | 700/150/150 | 200 | 304 | Boundary Condition |
| MOF. | 1800/360/360 | 200 | up to 350 | Boundary Condition |

of the two acting forces are randomized. The boundary nodes of the sheet are kept in place. From every material configuration, we take 70 simulations for training and 15 simulations for validation and testing respectively. For the out-of-distribution task, we only trained on Young's modulus ranging between 60 and 500, while testing on Young's modulus values 10, 30, 750, and 1000.

## D.2 DEFORMABLE PLATE

The original task was introduced in Linkerhägner et al. (2023), generated using Simulation Open Framework Architecture (SOFA) (Faure et al., 2012). It uses 3 different Poisson's ratios and 9 different trapezoidal meshes. We increase the difficulty of this dataset by introducing more complex initial starting conditions. This is done by selecting a random Poisson's ratio, simulating for 11 steps, and then switching to another Poisson's ratio. Then, the simulation continues for 39 steps. The first 11 steps are then discarded and step 12 is then the initial step for the dataset (and is referred to step 0 throughout the paper).

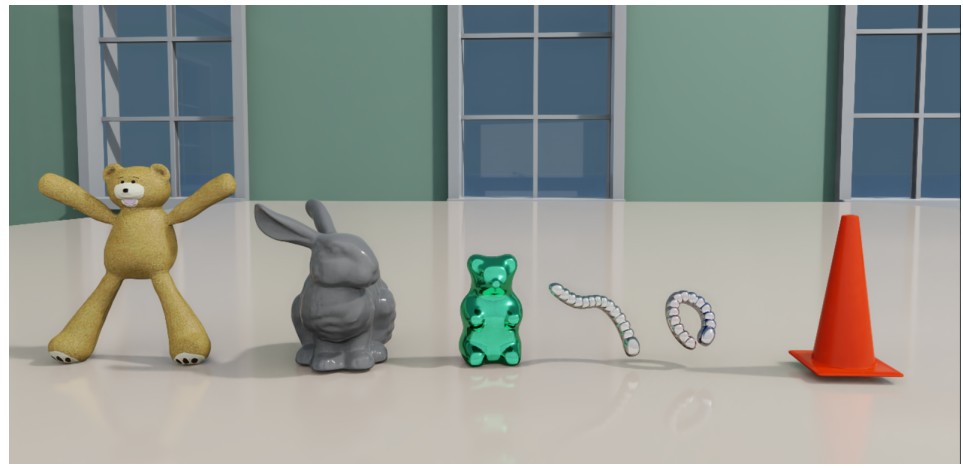

Figure 9: Six objects used in the *Mixed Objects Falling* task. From left to right: Teddy Bear, Bunny, Gummy Bear, Gummy Worm 1, Gummy Worm 2, and Traffic Cone.

### D.3 Tissue Manipulation

We use the original task introduced in Linkerhägner et al. (2023) without alterations. This dataset was also generated using SOFA (Faure et al., 2012).

### D.4 Falling Teddy Bear

Each trajectory of the dataset was created by choosing an angle from $[0°, 360°]$ for the first time step. To vary the material properties, we randomly select one possible combination of the Young's modulus and Poisson's ratio from the sets generated by `np.linspace(1 × 10^5, 1 × 10^6, 1000)` and `np.linspace(0.0, 0.499, 100)`, respectively. This dataset is generated using NVIDIA Isaac Sim (NVIDIA, 2022a), which utilizes PhysX 5.0 (NVIDIA, 2022b) to simulate tetrahedral meshes based on initial CAD models.

### D.5 Mixed Objects Falling

The simulation uses the setup from *Falling Teddy Bear*. In addition to the Teddy, we include other objects to encourage diversity. In total, there are six different objects presented in the dataset. We report an image of their high-resolution meshes in Figure 9. From every object, we use 300 simulations for the training split and 60 simulations for the test and validation split respectively.

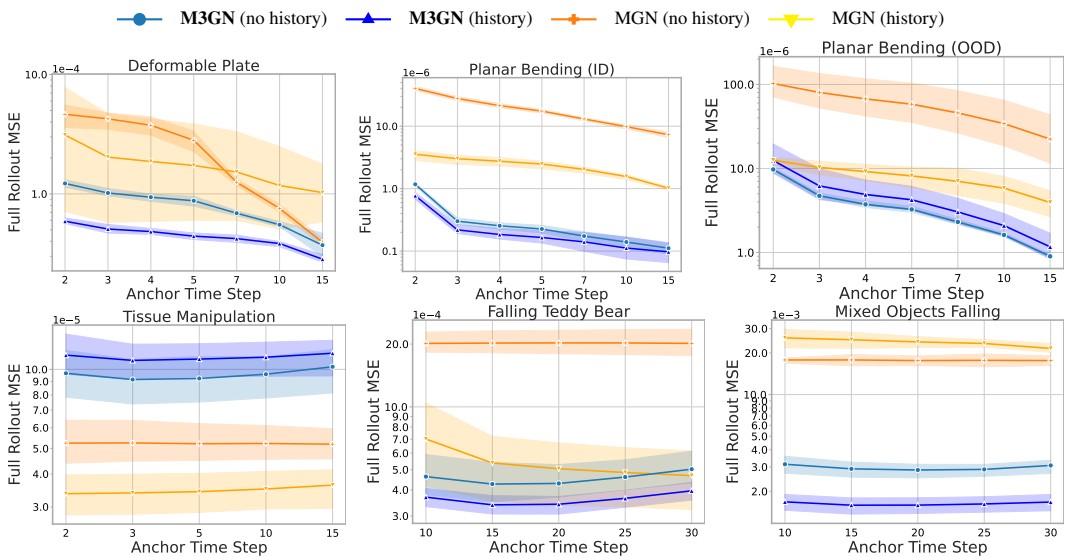

Figure 10: Log-scale MSE over full rollouts on the validation split for M3GN and MGN comparing history features. The better performing hyperparameter configuration was chosen for the final evaluation on the test dataset.

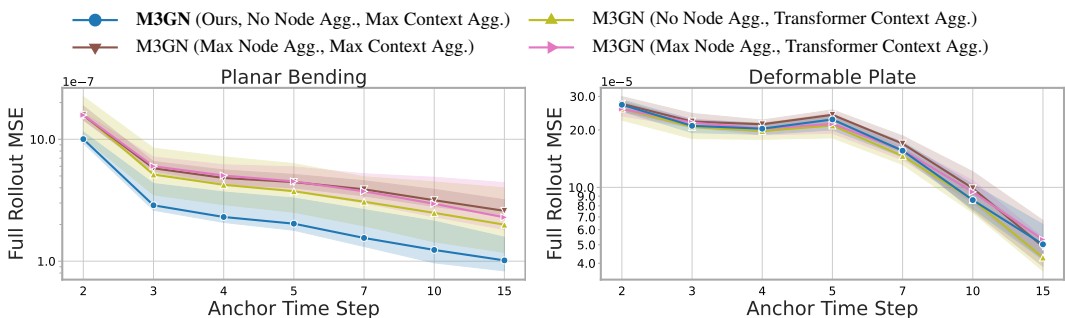

Figure 11: Log-scale MSE over full rollouts for the *Planar Bending* (**Left**) and *Deformable Plate* (**Right**) tasks for different context and node aggregation methods. The node-level maximum context aggregation of M3GN performs best for *Planar Bending*, while all methods work roughly equally well on the *Deformable Plate* task.

# E ADDITIONAL RESULTS

## E.1 HYPERPARAMETER OPTIMIZATION

We observed that the history inclusion of previous velocities has a big impact on the result of the simulation, depending on the task. To obtain optimal performance, we did an hyperparameter optimization on the validation split comparing history features. The results for M3GN andMGN are given in Figure 10.

## E.2 EVALUATIONS.

**Aggregation** Figure 11 shows results for different context aggregation schemes, comparing global and node-level aggregation, and a transformer-based aggregation over the context set to a simple maximum operator. The node-level maximum context aggregation of M3GN works best for *Planar Bending*. For *Deformable Plate*, there is no significant difference between node-level and global contexts, or between maximum and transformer-based aggregations.

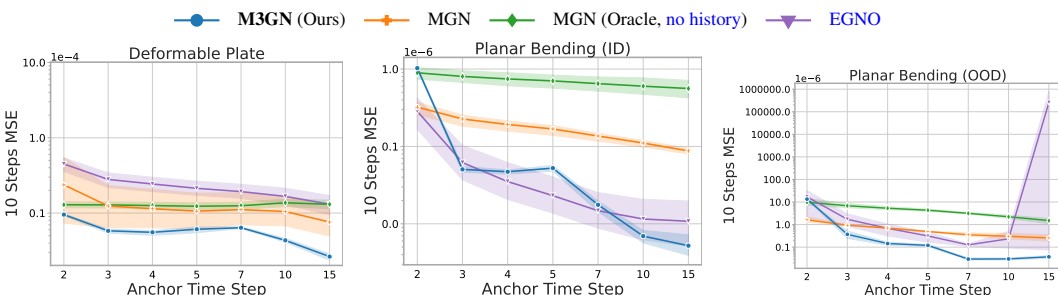

Figure 12: Log-scale MSE over the first 10 steps after the anchor time step for the *Deformable Plate* (**Left**) task, the *Planar Bending* (**Center**) task, and the *Planar Bending* task with out-of-distribution material properties for the test trajectories. M3GN steadily improves its performance when provided with additional context information and a later anchor time step. When evaluating this metric, we also outperform MGN Oracle on the *Deformable plate* task.

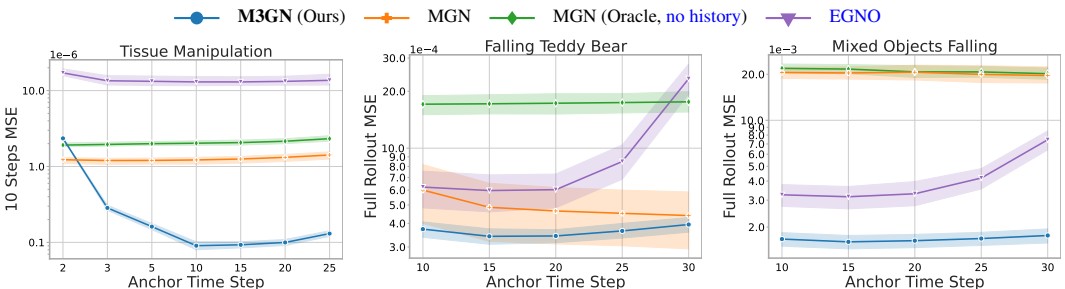

Figure 13: Log-scale MSE over the first 10 steps after the anchor time step for the *Tissue Manipulation* (**Left**), *Falling Teddy Bear* (**Middle**) and *Mixed Objects Falling* (**Right**) tasks for different methods. M3GN significantly outperforms all baselines on *Tissue Manipulation* when provided with a context size larger than 3. For the other tasks, M3GN significantly improves over both MGN variants, but does not benefit much from additional context information.

Figure 12 and Figure 13 show the performance of M3GN compared to the baselines evaluated on the 10 steps MSE. Here, instead of the whole trajectory, only the first 10 steps after the anchor time step are used to evalute the MSE.

**MSE over time** To gain better insights into the rollout stability of the model predictions, we report the Mean Squared Error (MSE) over timesteps in Figure 14, 15, 16, 17, 18, and Figure 19. Overall, our model M3GN demonstrates great robustness against error accumulation, benefiting from the inherent trajectory representation provided by the ProDMP method. MGN works well on the *Tissue Manipulation* task, but fails to incorporate the correct context information on other tasks. EGNO performs in general worse except on the *Planar Bending* tasks.

### E.3 VISUALIZATIONS.

In Figure 20, we include a latent space visualization of the Planar Bending task, where simulations with 9 different Young's Modulus values are clustered according to their material properties. The t-SNE projection of the 64-dimensional latent node vectors demonstrates clear clustering, indicating that the model effectively captures and differentiates material characteristics based on learned task representations.

We further provide additional visualizations for M3GN, MGN and MGN (Oracle) for exemplary simulations of all tasks. Each visualization shows the same simulated trajectory for different time steps (columns) and different methods (rows).

- Figure 21 shows a simulation of the *Planar Bending* task for a context set size of 5.
- Figure 22 visualizes the *Deformable Plate* task for a context set size of 6.

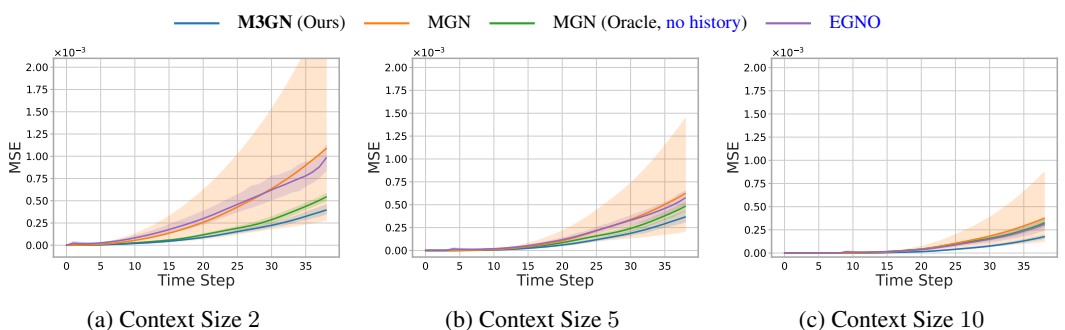

Figure 14: MSE over timesteps for the *Deformable Plate* task.

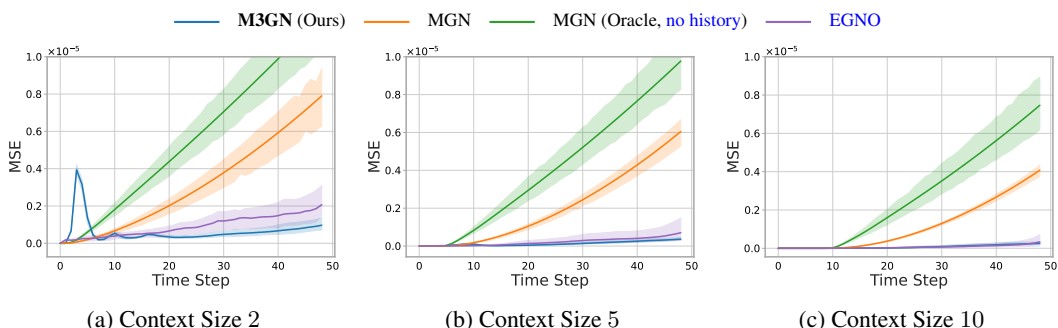

Figure 15: MSE over timesteps for the *Planar Bending (ID)* task.

- Figure 23 shows a *Tissue Manipulation* visualization for a context set size of 6.
- Figure 24 provides an examplary *Teddy Bear Falling* for a context set size of 20.
- Figure 25 and Figure 26 show two different simulated *Mixed Objects Falling* for a context set size of 20.

Across tasks, M3GN provides accurate simulations, whereas MGN, especially when not provided the additional material information as oracle knowledge, sometimes fails to respect the material properties or predicts a drift in the solution for later time steps.

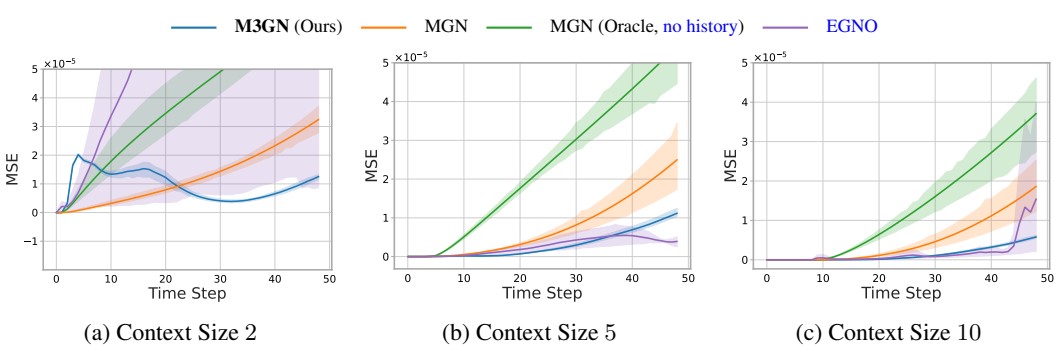

Figure 16: MSE over timesteps for the *Planar Bending (OOD)* task.

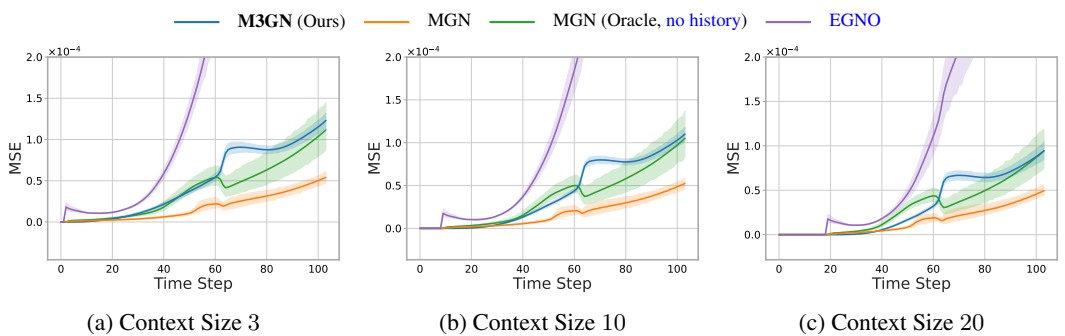

Figure 17: MSE over timesteps for the *Tissue Manipulation* task.

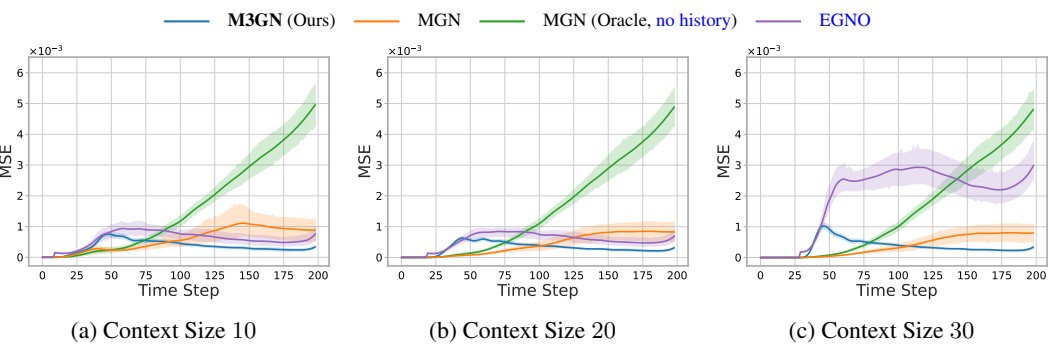

Figure 18: MSE over timesteps for the *Falling Teddy Bear* task.

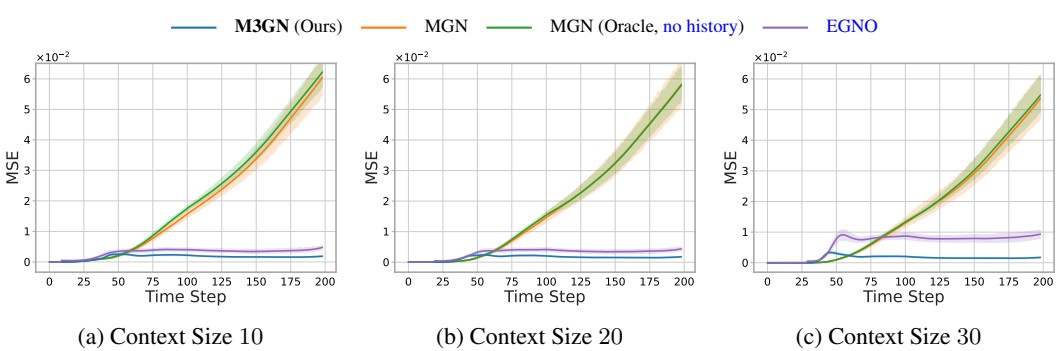

Figure 19: MSE over timesteps for the *Mixed Objects Falling* task.

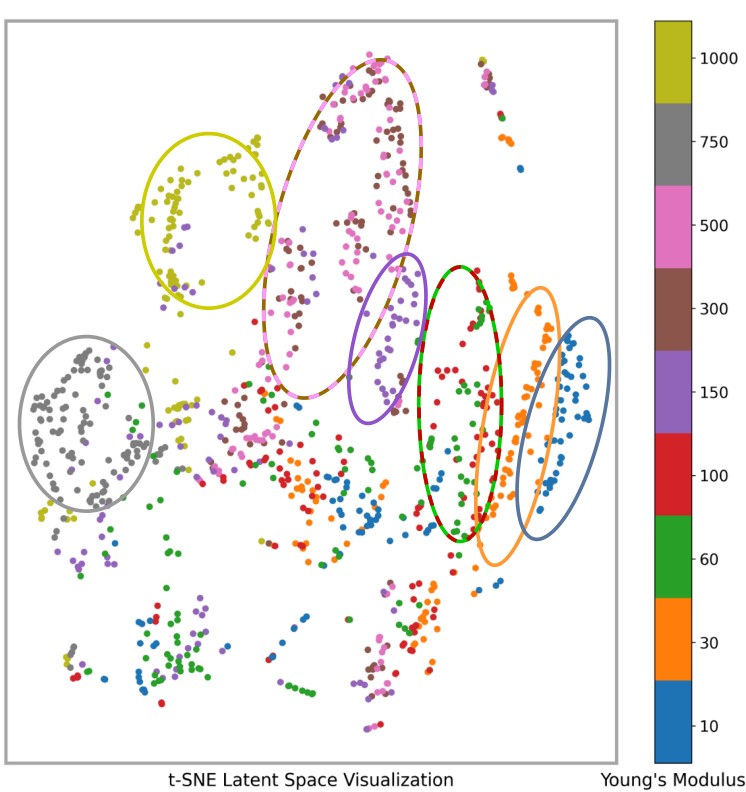

Figure 20: This figure shows a latent space visualization of the Planar Bending task for trajectories with 9 different Young's Modulus values, using a context size of 10. Each dot represents a 64-dimensional latent node vector projected to 2D using the t-SNE algorithm (van der Maaten & Hinton, 2008). Dots of the same color correspond to latent node descriptions for the same task, each simulated with a unique Young's Modulus. The visualization reveals distinct clustering in the latent space, with similar material properties grouped closer together, highlighting the relationship between material characteristics and the learned task representations. To improve clarity, points corresponding to nodes on the plate's edge were excluded, as their constant boundary condition resulted in unvarying latent descriptions.

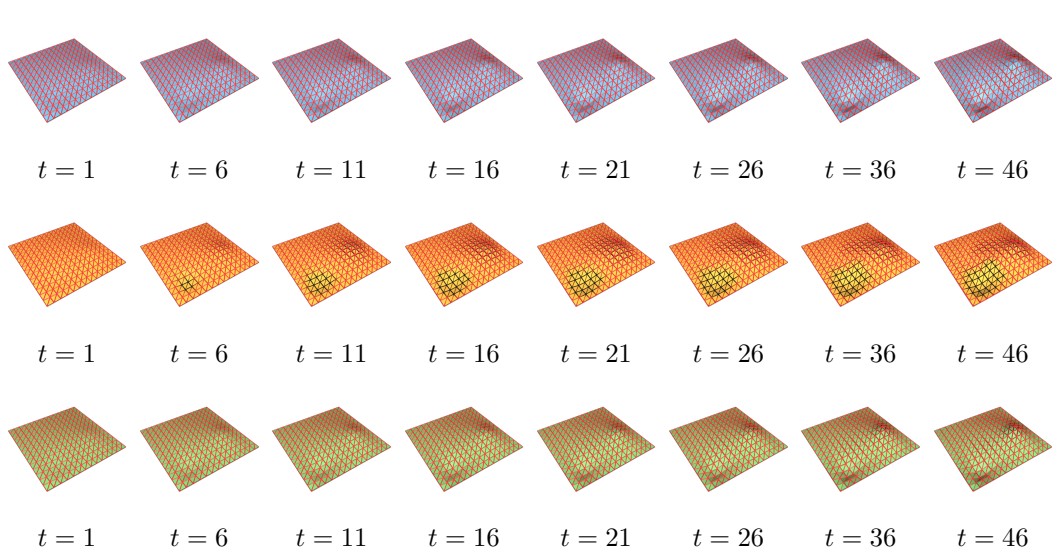

$t = 1$    $t = 6$    $t = 11$    $t = 16$    $t = 21$    $t = 26$    $t = 36$    $t = 46$

$t = 1$    $t = 6$    $t = 11$    $t = 16$    $t = 21$    $t = 26$    $t = 36$    $t = 46$

$t = 1$    $t = 6$    $t = 11$    $t = 16$    $t = 21$    $t = 26$    $t = 36$    $t = 46$

Figure 21: Simulation over time of an exemplary test trajectory from the **Planar Bending** task by M3GN, MGN, and MGN with oracle information. The **context set size** is set to 5. All visualizations show the colored **predicted mesh**, a **collider or floor**, and a **wireframe** of the ground-truth simulation. M3GN can accurately predict the correct material properties, resulting in a highly accurate simulation.

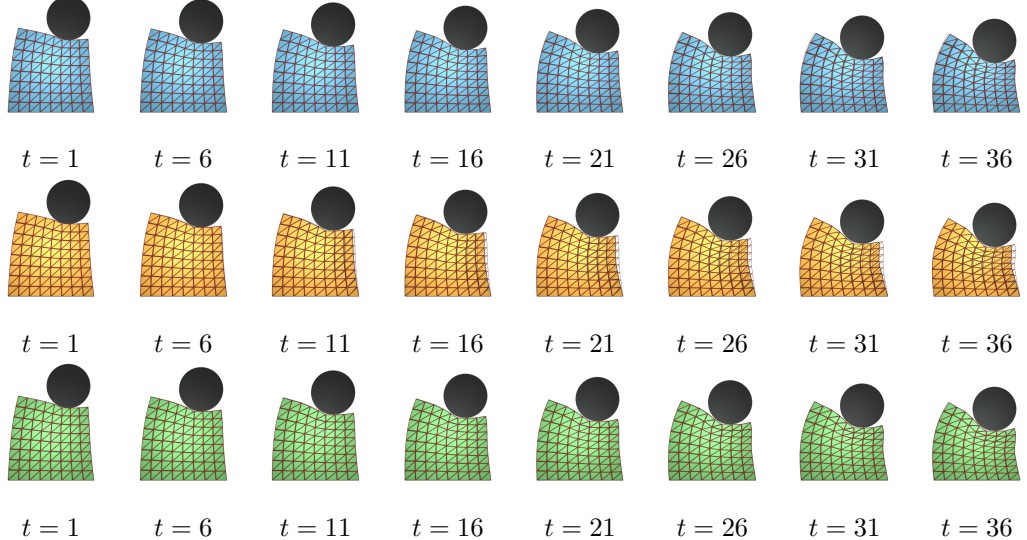

$t = 1$    $t = 6$    $t = 11$    $t = 16$    $t = 21$    $t = 26$    $t = 31$    $t = 36$

$t = 1$    $t = 6$    $t = 11$    $t = 16$    $t = 21$    $t = 26$    $t = 31$    $t = 36$

$t = 1$    $t = 6$    $t = 11$    $t = 16$    $t = 21$    $t = 26$    $t = 31$    $t = 36$

Figure 22: Simulation over time of an exemplary test trajectory from the **Deformable Plate** task by M3GN, MGN, and MGN with oracle information. The **context set size** is set to 6. All visualizations show the colored **predicted mesh**, a **collider or floor**, and a **wireframe** of the ground-truth simulation. M3GN can accurately predict the correct material properties, resulting in a highly accurate simulation.

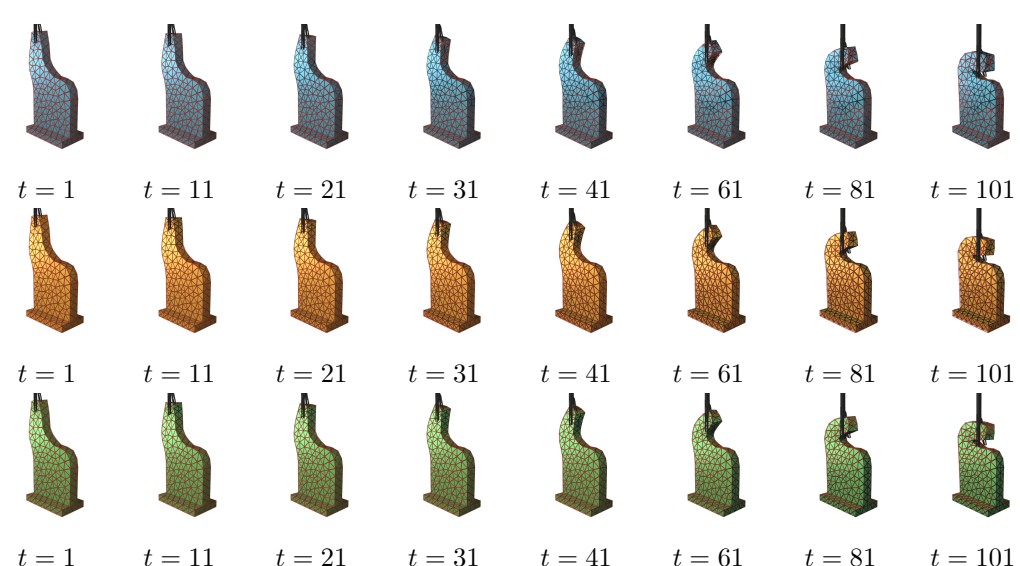

Figure 23: Simulation over time of an exemplary test trajectory from the **Tissue Manipulation** task by M3GN, MGN, and MGN with oracle information. The **context set size** is set to 6. All visualizations show the colored **predicted mesh**, a **collider or floor**, and a **wireframe** of the ground-truth simulation. All methods can solve the task, however MGN is drifting a tiny bit to the left over time.

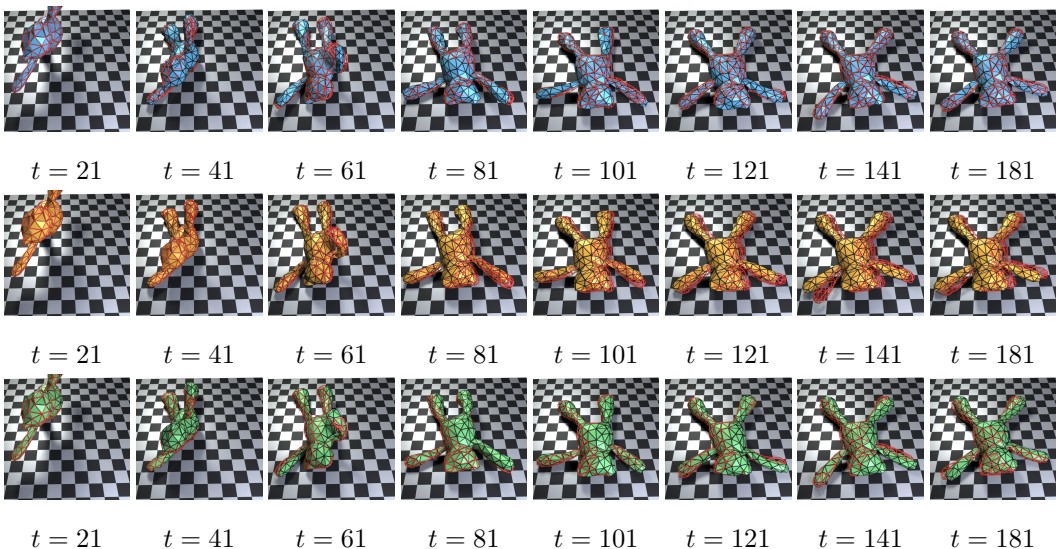

Figure 24: Simulation over time of an exemplary test trajectory from the **Falling Teddy Bear** task by M3GN, MGN, and MGN with oracle information. The **context set size** is set to 20. All visualizations show the colored **predicted mesh**, a **collider or floor**, and a **wireframe** of the ground-truth simulation. M3GN significantly outperforms both step-based baselines.

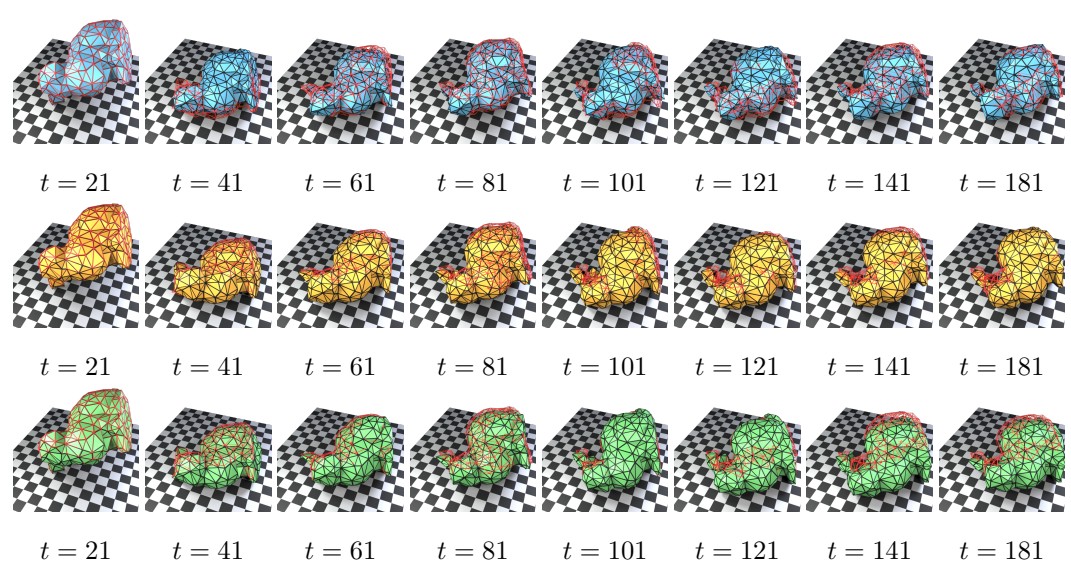

Figure 25: Simulation over time of a bunny from the **Mixed Objects Fall** task by M3GN, MGN, and MGN with oracle information. The **context set size** is set to 20. All visualizations show the colored **predicted mesh**, a **collider or floor**, and a **wireframe** of the ground-truth simulation. M3GN significantly outperforms both step-based baselines.

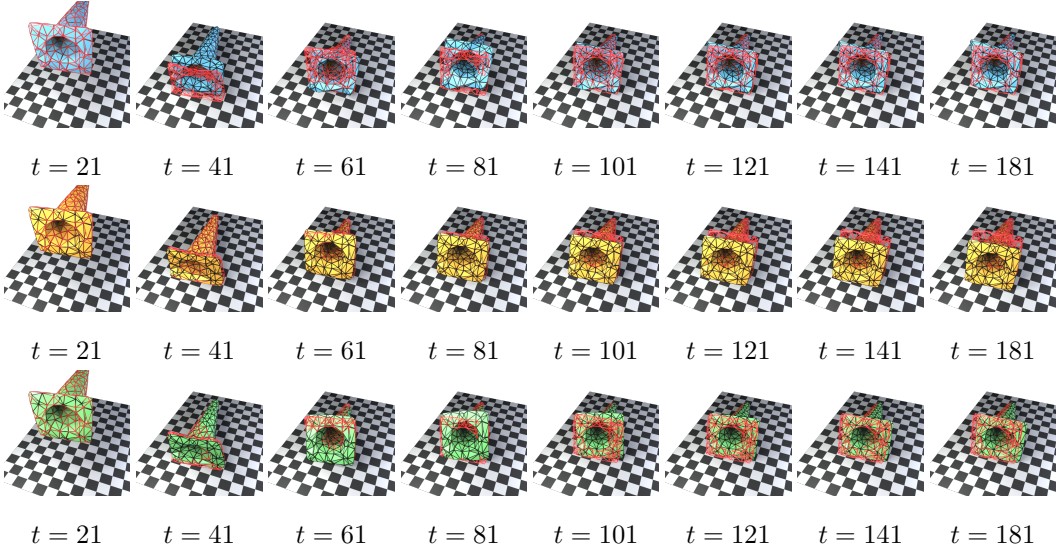

Figure 26: Simulation over time of a pylon from the **Mixed Objects Fall** task by M3GN, MGN, and MGN with oracle information. The **context set size** is set to 20. All visualizations show the colored **predicted mesh**, a **collider or floor**, and a **wireframe** of the ground-truth simulation. M3GN significantly outperforms both step-based baselines. The simulation generated by MGN is severly affected by drift.

