# OpenReview forum: "Latent Task-Specific Graph Network Simulators"
_ICLR.cc/2025/Conference — Submitted to ICLR 2025_

### Official Review · Reviewer_fGhd · 2024-10-30

**Soundness:** 3
**Presentation:** 4
**Contribution:** 3
**Rating:** 8
**Confidence:** 5

**Summary:**

This paper first propose a meta-learning framework to efficiently learn generalizable mesh-based dynamic prediction tasks. Different from previous graph neural simulators which predict the state updates in a step-by-step manner, the proposed M3GN targets to predict the whole trajectories by a conditional neural process to effectively diminish the error accumulation issue.

**Strengths:**

Strength:
1. Adopting meta-learning to deal with dynamic prediction tasks is novel, especially the concept of regarding each tajectory as a new task is interesting.

2. The authors consider past information and the eventual state of the collider as the condition to predict the subsequent movement trajectory, which make the network infer the future from the past rather than remember the dynamic behaviour of a certain material. In addition, predicting the whole rest path by a single forward pass could significantly improve the efficiency, compared with previous Graph-based single timestep prediction.

**Weaknesses:**

Weakness:
1. This paper is highly related to the Graph-based Neural Simulators. However, in the related work section, the latest advancements in this field are not included, and most of the work discussed is from 2023 or earlier. This could make the paper appear somewhat outdated. I believe this section could benefit from a more comprehensive overview of the field, especially more works from 2024. Below are two of the latest advancements about Graph Network Simulators that I recommend the authors to discuss them in Section 2.1 ,or better, use them as baselines for comparison. However, given the tight rebuttal timeline, it is also tolerated that concurrent works were not included for comparison.

    (1) "DEL: Discrete Element Learner for Learning 3D Particle Dynamics with Neural Rendering" 2024   ..  This work integrate traditional Newton mechanics into the graph network design to benefit from mechanics priors for longer term prediction.

    (2) "Equivariant graph neural operator for modeling 3d dynamics" 2024  ..  This paper deal with dynamic prediction tasks as trajectory-level rather than next-step level by operator learning, which is somewhat relavent with this reviewing work. Also, it handle the equivariant issues.

2. For Equation 3, does it use past trajectory collider states when encoding z because I saw that you seem to only use the latest state, or does it rely solely on the historical information of the deformed object? I believe it would be more reasonable to use all the historical information of the collider here as well, since the deformation of the mesh is passive.

3. If this method is trained on an elastic dataset, can it generalize directly to elastoplastic materials? I believe it would be worthwhile to discuss the generalization across different materials in the experiments, rather than limiting it to variations in mechanical parameters within the same material.

4. Line 276 mentions that the context information z is concatenated with the node features. Is the same z concatenated to each node?

5. Finally, the neural network predicts a set of weights, and the shape of the weight matrix is 𝑇, 𝐷,3. Which basis functions are these weights applied to in order to obtain the predicted trajectory? Are they precomputed from the historical trajectory? If yes, how?

In appendix A.2 "Initially, we integrate a relative goal position as part of the node weights w" What's the exact mean of the relative goal position?

I will raise the score if most of concerns are well addressed by the authors.

**Questions:**

See above.

---

> ### Author Response · Authors · 2024-11-24
> **Rebuttal [1/2]**
>
> We thank the reviewer for their constructive feedback and thoughtful suggestions. This rebuttal addresses the inclusion of recent advancements in the related work and as a baseline comparison, and clarifies design choices such as the use of collider states and context information.  Additional details on architectural components and trajectory prediction mechanics are also provided to enhance clarity and completeness.
>
> ### Responses to Specific Concerns
>
> > 1. This paper is highly related to the Graph-based Neural Simulators. However, in the related work section, the latest advancements in this field are not included, and most of the work discussed is from 2023 or earlier. This could make the paper appear somewhat outdated. I believe this section could benefit from a more comprehensive overview of the field, especially more works from 2024. Below are two of the latest advancements about Graph Network Simulators that I recommend the authors to discuss them in Section 2.1 ,or better, use them as baselines for comparison. However, given the tight rebuttal timeline, it is also tolerated that concurrent works were not included for comparison.
> (1) "DEL: Discrete Element Learner for Learning 3D Particle Dynamics with Neural Rendering" 2024 .. This work integrate traditional Newton mechanics into the graph network design to benefit from mechanics priors for longer term prediction.
> (2) "Equivariant graph neural operator for modeling 3d dynamics" 2024 .. This paper deal with dynamic prediction tasks as trajectory-level rather than next-step level by operator learning, which is somewhat relavent with this reviewing work. Also, it handle the equivariant issues.
>
> We thank the reviewer for highlighting the importance of including recent advancements in Graph Neural Simulators. The related work section has been updated to discuss both DEL: Discrete Element Learner for Learning 3D Particle Dynamics with Neural Rendering and Equivariant Graph Neural Operator for Modeling 3D Dynamics. EGNO has also been implemented as a baseline, but due to the short rebuttal timeline, its evaluation is not fully complete and will be included in the final revision on Wednesday. Additionally, we discuss AURORA, a foundation model approach for climate predictions,  as an alternative to meta-learning.
>
> > 2. For Equation 3, does it use past trajectory collider states when encoding z because I saw that you seem to only use the latest state, or does it rely solely on the historical information of the deformed object? I believe it would be more reasonable to use all the historical information of the collider here as well, since the deformation of the mesh is passive.
>
> We thank the reviewer for pointing out this potential ambiguity. The model does incorporate the historical trajectory of the collider when encoding z, ensuring that both the past collider states and the deformation history of the object are considered. The paper has been updated to clarify this aspect.
>
> > 3. If this method is trained on an elastic dataset, can it generalize directly to elastoplastic materials? I believe it would be worthwhile to discuss the generalization across different materials in the experiments, rather than limiting it to variations in mechanical parameters within the same material.
>
> The proposed method has not been tested on elastoplastic materials, as this setup was not included in our current data suite. However, we consider this an interesting direction for future research. If the tasks share common aspects and the context sufficiently captures the relevant properties of the new material, the model should, in principle, be able to generalize across different material types. This hypothesis aligns with the adaptability demonstrated by the method in other scenarios: to evaluate the model's generalization capabilities, we created a new data split in the Planar Bending task. Here, the Young's Modulus values used for training ranged between 60 and 500, while the test set included Young's Modulus values from [10, 30, 750, 1000], representing a clearly out-of-distribution scenario. Results from these experiments, included in the appendix, show that M3GN significantly outperforms both MGN and MGN (Oracle) in this setting, demonstrating strong generalization to out-of-distribution material properties. The reviewer can find the results in Figure 5.

---

> > ### Author Response · Authors · 2024-11-24
> > **Rebuttal [2/2]**
> >
> > > 4. Line 276 mentions that the context information z is concatenated with the node features. Is the same z concatenated to each node?
> >
> > We appreciate the opportunity to address the reviewer’s question. The context information z is not the same for each node. A core design choice of M3GN is the use of per-node latent features z_v​, which are directly output by the context encoder. As shown in the ablation studies in Figure 8, using a global aggregation instead of per-node latent features resulted in worse performance. We suspect this is because material properties can also be processed locally, making per-node features more effective. Additionally, we have added a latent space visualization (see Appendix D, Figure 17) to demonstrate that while latent features differ across nodes in the mesh, they still cluster together for the same trajectory and are distinct for simulations with different material properties.
> >
> > > 5. Finally, the neural network predicts a set of weights, and the shape of the weight matrix is 𝑇, 𝐷,3. Which basis functions are these weights applied to in order to obtain the predicted trajectory? Are they precomputed from the historical trajectory? If yes, how?
> >
> > We appreciate the opportunity to clarify this point. To address the question, the basis functions used are the positional basis functions from the ProDMP method, which are derived by solving the underlying ODE of DMPs. These functions act as an inductive bias and can be precomputed offline, remaining constant during training and inference. They are not derived from the historical trajectory but rather form a fixed part of the model's design.
> >
> > To improve clarity, we have added a detailed explanation of the MP method in Appendix A, along with visual illustrations of the used basis functions in Figure 10 (b) and (c). This new section provides a comprehensive overview of how the basis functions operate within the framework.
> >
> > > In appendix A.2 "Initially, we integrate a relative goal position as part of the node weights w" What's the exact mean of the relative goal position?
> >
> > We agree that the explanation was insufficiently detailed. Below is a brief reply, which has also been added to our Appendix A.3, along with a detailed presentation of the MP approaches.
> >
> > We use ProDMP as our trajectory generator, which models the trajectory as a dynamic system. The dynamic system includes a goal attractor that represents the asymptotic convergence point as $t \rightarrow \infty$. By default, this goal term is defined in absolute coordinates. However, it can also be modeled relative to the initial position of the trajectory. In this case, the relative goal $g_{\text{rel}}$ is predicted, and its absolute counterpart is calculated as $g_{\text{abs}} = g_{\text{rel}} + y_b$. This approach is particularly useful for predicting the goal in the coordinate system relative to a node’s starting position. Since we aim to achieve a translation-equivariant approach (where absolute node positions are encoded as relative edge features between nodes), predicting relative goal positions aligns well with this design principle.
> >
> > ### Concluding Remarks
> >
> > We appreciate the reviewer’s valuable feedback, which has helped us improve the paper’s clarity, completeness, and experimental evaluation. With the revisions and additional experiments addressing the all of the concerns, we believe the updated manuscript better highlights the strengths and contributions of our approach. We hope the changes meet the reviewer’s expectations and demonstrate the merit of our work.

---

> > > ### Comment · Reviewer_fGhd · 2024-11-25
> > > **Response to the Feedback of the authors**
> > >
> > > Thanks for the author's reply. I had some misconceptions about this method earlier, and the author's reply dispelling most of my concerns. Moreover, the content of the revised manuscript is richer and the quality has been improved. Therefore, I decide to raise my score.

---

> > > > ### Author Response · Authors · 2024-11-25
> > > >
> > > > We thank the reviewer for their thoughtful reconsideration and for raising the score. We're glad the clarifications and updates addressed the earlier concerns and that the improvements to the manuscript have been well-received. The feedback provided has been very important in refining the paper, and we greatly appreciate the constructive input.

---

### Official Review · Reviewer_iEk9 · 2024-11-03

**Soundness:** 3
**Presentation:** 3
**Contribution:** 3
**Rating:** 6
**Confidence:** 4

**Summary:**

In this paper, the authors propose a graph network simulator that combines movement primitives and trajectory-level meta-learning. The network uses the simulation history as the context information to predict the deformation for objects with unknown properties. They also use probabilistic dynamic movement primitives to represent the future trajecteries and directly predicts the full simulation trajectories instead of iteratively predicting the next-step. Experiments show that it outperforms STOA in different simulation tasks. Abalation studies validate the effectivenss of the design choice.

**Strengths:**

This work aims to address two important problems in learning-based simulation:

1. It treats the simulation as a trajectory-level meta-learing problem and use trajectory history as the context to predict future trajectories.

2. It mitigates the problem of error accumulation by using ProDMP to directly predict the full simulation trajectories.

The paper is well structured and written.

**Weaknesses:**

1. Some descriptions are unclear and some important details are missing.
(1) in line 242, "graph edges between the deformable object and collider are added based on physical proximity to model interactions
 between objects." what is the physical proximity exactly? Since the deformation mesh node position for the end timestep is unknown, I suppose we cannot use that to compute the distance. Whether this edge creation is done only for known timesteps or if it's updated during prediction?

(2) in line 231, why is the term c_1y_1(t) + c_2y_2(t) only depending on the inital conditions? What is the representation of the pre-computed basis fuction \phi?

2. More detailed description of the training/val/test split should be added. Specify how trajectories are divided between training, validation, and test sets. What are different between training and test? Clarify if test trajectories involve different objects, material properties, or initial conditions than training trajectories. In the limitation part, it is claimed ''We currently consider each trajectory as a task, and require initial states of this trajectory as a context set during inference."

3. Since the method needs a trajectory with simulated states as context, the author better include a runtime comparison between your method (including context computation) and traditional simulators for predicting the same number of future timesteps and discuss the trade-offs between computation time and accuracy compared to traditional simulators.

**Questions:**

1. What is the timestep for simulation?

2. A figure illustraing all the relation and symbols of input, output can be added. Fig.3 Right is not information for undertanding the task setting.

---

> ### Author Response · Authors · 2024-11-24
> **Rebuttal [1/2]**
>
> We thank the reviewer for their thoughtful feedback and constructive comments. The points raised have helped us identify areas for clarification and improvement, which we have addressed in the revised manuscript. In this rebuttal, we provide detailed responses to the specific concerns, including clarifications on the physical proximity used for edge creation, the basis function representation, training/test splits, and further elaboration on runtime comparisons. We hope the revisions will provide a clearer understanding of the methodology and enhance the overall quality of the paper.
>
> ### Responses to Specific Concerns
>
> >Some descriptions are unclear and some important details are missing. (1) in line 242, "graph edges between the deformable object and collider are added based on physical proximity to model interactions between objects." what is the physical proximity exactly? Since the deformation mesh node position for the end timestep is unknown, I suppose we cannot use that to compute the distance. Whether this edge creation is done only for known timesteps or if it's updated during prediction?
>
> We thank the reviewer for highlighting this point. The "physical proximity" refers to the creation of an edge between the deformable object and collider when the distance between the two is smaller than a given threshold of 0.3. We updated the hyperparameter section in Appendix C to include this value for clarity. This edge creation process is applied only to the context. During prediction, we rely on the anchor step and the previous context to predict the entire remaining trajectory. Therefore, no information about the proximity of future steps is required when unrolling the ProDMP trajectories.
>
> > (2) in line 231, why is the term c_1y_1(t) + c_2y_2(t) only depending on the inital conditions? What is the representation of the pre-computed basis fuction \phi?
>
> We briefly reply to this question in below, while added a detailed presentation of the ProDMP theory in Appendix A.
>
> ProDMP, as a parameterized trajectory generator, models a trajectory using a second-order dynamical system. This system is governed by a second-order linear ordinary differential equation (ODE). ProDMP builds upon its predecessor, DMP, which computes the trajectory by applying numerical integration from the start to the end of the trajectory. In contrast, ProDMP directly computes the closed-form solution of the second-order ODE as the position trajectory. This closed-form solution involves two coefficients, $c_1$ and $c_2$, corresponding to the complementary functions. From the fundamentals of solving linear ODEs, these coefficients can be uniquely determined given two initial conditions. In the Appendix A, Equation 19, we present the form of their solutions.
>
> Regarding the basis functions, ProDMPs identify reusable terms, specifically the position and velocity basis functions, denoted by \Phi(t) and \dot{\Phi}(t), respectively. These are visualized in Fig. 10b in Appendix A. The mathematical representation is discussed in Equation 12 and Equation 14-16 of Appendix A.
>
> > More detailed description of the training/val/test split should be added. Specify how trajectories are divided between training, validation, and test sets. What are different between training and test? Clarify if test trajectories involve different objects, material properties, or initial conditions than training trajectories. In the limitation part, it is claimed ''We currently consider each trajectory as a task, and require initial states of this trajectory as a context set during inference.”
>
> We thank the reviewer for the valuable feedback. We have updated Appendix D to provide more details about the training, validation, and test split. For the test splits, most tasks so far have involved in-distribution data, with variations in starting positions and collider trajectories, but with material properties either identical to or interpolated from those used in training.
>
> To evaluate the model's generalization capabilities, we created a new data split in the Planar Bending task. In this split, the Young's Modulus values used for training ranged between 60 and 500, while the test set included values from [10, 30, 750, 1000], representing a clearly out-of-distribution scenario. Results from these experiments, included in the appendix, demonstrate that M3GN significantly outperforms both MGN and MGN (Oracle) in this setting, indicating strong generalization to out-of-distribution material properties.

---

> > ### Author Response · Authors · 2024-11-24
> > **Rebuttal [2/2]**
> >
> > >  Since the method needs a trajectory with simulated states as context, the author better include a runtime comparison between your method (including context computation) and traditional simulators for predicting the same number of future timesteps and discuss the trade-offs between computation time and accuracy compared to traditional simulators.
> >
> > We appreciate the reviewer's suggestion. We have included a comparison of computation time for traditional simulators in the revised manuscript in Figure 8. Depending on the simulator and task, our method achieves up to a 400x speedup over traditional simulators, demonstrating significant efficiency gains. This trade-off between computation time and accuracy is discussed in more detail, highlighting the benefits of using M3GN for faster predictions while maintaining strong accuracy in simulation tasks.
> >
> > ### Concluding Remarks
> >
> > We would like to thank the reviewer again for their thorough and thoughtful feedback, which has been instrumental in improving the clarity and depth of our manuscript. We believe the revisions, including the added details on edge creation, the ProDMP theory, training/test splits, and runtime comparisons, provide a more comprehensive understanding of the methodology and its performance. We hope the updated manuscript addresses all the reviewer’s concerns and enhances the overall contribution of our work.

---

> > > ### Author Response · Authors · 2024-12-02
> > >
> > > We hope this response addresses the reviewer’s concerns. Should the reviewer have any additional questions or require further clarification, we would be happy to provide further details.

---

### Official Review · Reviewer_roSh · 2024-11-04

**Soundness:** 2
**Presentation:** 2
**Contribution:** 2
**Rating:** 3
**Confidence:** 4

**Summary:**

This paper introduces Movement-primitive Meta-MeshGraphNet (M3GN), a model for simulating object deformations in data-limited scenarios. M3GN combines meta-learning and movement primitives to improve the adaptability and accuracy of Graph Network Simulators (GNSs) by framing mesh-based simulation as a meta-learning task.

**Strengths:**

The paper takes a novel approach to enhancing rollout stability by predicting entire future mesh states, and it incorporates a meta-learning scheme to improve adaptability within the simulation framework.

**Weaknesses:**

While the approach appears novel, the rationale behind certain modules in the model is unclear, and the results do not provide sufficient evidence to justify their inclusion. Also, the paper is not clearly written and sometimes hard to follow. The detailed comments and suggestions are listed below.

**Questions:**

1. The model's architecture is not clearly explained, and it is unclear why certain modules are necessary. For example, from the results, it seems that MGN, even without history information, can surpass M3GN in performance. This raises questions about the value of incorporating historical information in M3GN. Moreover, the experimental results do not clearly demonstrate the necessity or advantages of using a meta-learning scheme. A thorough analysis on how meta-learning benefits model performance would be valuable, including ablation studies comparing model performance with and without meta-learning.
2. The authors claim that the baseline MGN does not incorporate historical information, which appears inaccurate. In certain datasets, MGN does include history. For a fair comparison, the MGN baseline should also be evaluated with historical data to assess its impact on performance.
3. The results section only reports the average MSE across all time steps. It would be helpful to provide a comparison of MSE over the number of prediction steps, as this would give insight into the model's performance stability over time as claimed in the paper.
4. Based on Figure 3, the proposed M3GN method does not appear to use ground truth collider information. If this is the case, does the collider state being predicted by the mode? How accurate is the collider state prediction, especially when history steps are limited? Additionally, including collider ground truth (as in MGN) is actually intuitive and makes sense, as the primary goal of developing a simulation model is to understand how a solid deforms under varying contact forces and obstacle displacements. Predicting these external forces may not be necessary for achieving this objective.
5. It would be informative to visualize the node-level latent task descriptions learned by the model. Such visualizations could help in understanding how task-specific information is represented.
6. The datasets used in this paper have relatively small node counts compared to those in previous MGN studies or those used in other related papers. When the number of nodes increases significantly, it is concerned that M3GN may struggle due to the large number of historical steps required. Comparing M3GN’s memory usage with MGN’s would provide a more comprehensive evaluation.
7. The authors consider each trajectory as a separate task with varying context sizes. However, this approach may not align with the broader goals of meta-learning, as tasks are typically defined by consistent properties such as the same material setting. Currently, the meta-learning setup seems more focused on adapting to different context sizes rather than generalizing across diverse tasks.
8. As the input context size changes, will the number of predicted steps vary as well? If so, the model’s ability to generalize to different context sizes is unclear, and it may not be as flexible as MGN in this respect. Any experiments or evaluation on this aspect? Additionally, splitting single data points into multiple input-output sets seem to increase the effective amount of training data for M3GN, potentially creating an unfair comparison with MGN which use less training data.
9. The authors do not specify how material properties are incorporated. Also, it is unclear whether the test data involve material properties that are in-distribution or out-of-distribution relative to the training data. Providing this information is crucial for evaluating the model's generalization capabilities.
10. The authors mention that material node features are not added to M3GN. Given that these features enhance MGN's performance, it would be useful to understand the rationale for this exclusion and perform related ablation study.
11. Although the authors mention other methods in related work besides MGN, these methods are not included in the baselines. Some of these methods have better accuracy and efficiency. Including these additional baselines would provide a clearer view of M3GN’s comparative performance.
12. Will the data used in this study be publicly available? Making the dataset accessible would facilitate further research and replication studies.

---

> ### Author Response · Authors · 2024-11-24
> **Rebuttal [1/3]**
>
> We thank the reviewer for their valuable insights and detailed suggestions, especially in relation to textual clarifications and the inclusion of additional evaluations. We ran further trainings and evaluations regarding additional baselines and out-of-distribution testing. Additionally, we revised the paper to clarify the methodology and to provide additional visualizations to cover all requested issues.
> We provide a detailed response to the individual concerns raised by the reviewer.
> ### Responses to Specific Concerns
> > 1. The model's architecture is not clearly explained, and it is unclear why certain modules are necessary. For example, from the results, it seems that MGN, even without history information, can surpass M3GN in performance. This raises questions about the value of incorporating historical information in M3GN. Moreover, the experimental results do not clearly demonstrate the necessity or advantages of using a meta-learning scheme. A thorough analysis on how meta-learning benefits model performance would be valuable, including ablation studies comparing model performance with and without meta-learning.
>
> We thank the reviewer for the feedback and provided additional explanations to the model’s architecture in the revised paper. The historical information lets our method build an accurate latent belief of the object dynamics for each meta-task, allowing for a more accurate simulation. This is shown in Figure 7, where various meta-learning methods outperform methods which do not incorporate the historical context. The ablation in Figure 7 also shows that the combination of meta-learning and parameterizing the trajectory with movement primitives is vital for exact simulations. We would like to highlight that the MGN baseline is only better on small context sizes for the Tissue Manipulation experiment. On all other experiments and context sizes, M3GN clearly outperforms MGN.
>
> > 2. The authors claim that the baseline MGN does not incorporate historical information, which appears inaccurate. In certain datasets, MGN does include history. For a fair comparison, the MGN baseline should also be evaluated with historical data to assess its impact on performance.
>
> We follow the original MGN paper [1] for our baseline implementation. The paper states, e.g., in the description of Figure 5 that “limiting history size increases accuracy by preventing overfitting”, hence the reason why we have not considered it. To obtain a complete view and to follow the request of the reviewer, we are running the experiments for the MGN with history information and expect the results on Wednesday. We will update the paper accordingly.
>
> > 3. The results section only reports the average MSE across all time steps. It would be helpful to provide a comparison of MSE over the number of prediction steps, as this would give insight into the model's performance stability over time as claimed in the paper.
>
> We thank the reviewer for the suggestion. To address this, we now include Figure 13 and Figure 14 in the Appendix, reporting the MSE over timesteps for all the tasks. These figures provide a detailed comparison of the models' rollout stability across all timesteps. As shown, autoregressive methods like MGN and MGN (Oracle) suffer from error accumulation, leading to higher MSE as the time progresses. In contrast, our M3GN demonstrates significantly better stability, with much lower MSE across timesteps due to the trajectory representation leveraged by the ProDMP method. This supports our claim regarding the improved stability of our approach.
>
> > 4. Based on Figure 3, the proposed M3GN method does not appear to use ground truth collider information. If this is the case, does the collider state being predicted by the mode? How accurate is the collider state prediction, especially when history steps are limited? Additionally, including collider ground truth (as in MGN) is actually intuitive and makes sense, as the primary goal of developing a simulation model is to understand how a solid deforms under varying contact forces and obstacle displacements. Predicting these external forces may not be necessary for achieving this objective.
>
> We thank the reviewer for their thoughtful question. The proposed M3GN method does not predict collider states. Instead, we use the collider trajectory during context timesteps and include only the last future collider position as a feature during prediction. This approach has proven effective for all tasks in our experiments, enabling accurate deformation predictions without unnecessary architectural complexity.
>
> In preliminary experiments, we explored incorporating additional future collider positions, but found that it did not result in improved performance. However, we acknowledge that other tasks with different dynamics might require more extensive use of collider information. We addressed this in the paper, discussing how M3GN can adapt to incorporate additional collider states if needed.

---

> > ### Author Response · Authors · 2024-11-24
> > **Rebuttal [2/3]**
> >
> > > 5. It would be informative to visualize the node-level latent task descriptions learned by the model. Such visualizations could help in understanding how task-specific information is represented.
> >
> > We thank the reviewer for their suggestion. We added the visualization  of the node-level latent task descriptions for the Planar Bending task in Appendix E, Figure 15 to better understand how task-specific information is represented.
> >
> > The figure shows a latent space visualization for trajectories with 9 different Young's Modulus values, using a context size of 10. Each dot represents a 64-dimensional latent node vector projected to 2D using the t-SNE algorithm[2]. Dots of the same color correspond to latent node descriptions for the same task, each simulated with a unique Young's Modulus. The visualization reveals distinct clustering in the latent space, with similar material properties grouped closer together, highlighting the relationship between material characteristics and the learned task representations. To improve clarity, points corresponding to nodes on the plate's edge were excluded, as their constant boundary condition resulted in unvarying latent descriptions.
> >
> > > 6. The datasets used in this paper have relatively small node counts compared to those in previous MGN studies or those used in other related papers. When the number of nodes increases significantly, it is concerned that M3GN may struggle due to the large number of historical
> > steps required. Comparing M3GN’s memory usage with MGN’s would provide a more comprehensive evaluation.
> >
> > We thank the reviewer for the interesting remark. M3GN computes a latent task descriptor per context time step as shown in Equation 3 in the paper. It then aggregates over these time steps, using, e.g., the maximum of all steps. This aggregation can either happen in parallel over a batch of steps, or, if memory is a concern, sequentially. In the latter case, the final latent task descriptors are  updated step by step, allowing for a memory usage that is independent of the context size and thus comparable to that of MGN.
> >
> > While parallel context processing in M3GN requires more memory, we mitigate this during training by using a larger batch size for MGN, ensuring a fair comparison with similar memory usage across both methods. We updated the Appendix C to describe the memory usage in more detail.
> >
> > > 7. The authors consider each trajectory as a separate task with varying context sizes. However, this approach may not align with the broader goals of meta-learning, as tasks are typically defined by consistent properties such as the same material setting. Currently, the meta-learning setup seems more focused on adapting to different context sizes rather than generalizing across diverse tasks.
> >
> > We thank the reviewer for their insightful comment. We agree that meta-learning typically involves tasks with consistent properties, such as material settings. Combining multiple trajectories into a single task and using a small number of trajectories as a context set is an avenue we plan to explore in future work. Our results shows that even small context sizes of few simulation steps can describe various task properties and help to adapt the remaining simulation.
> >
> > > 8. As the input context size changes, will the number of predicted steps vary as well? If so, the model’s ability to generalize to different context sizes is unclear, and it may not be as flexible as MGN in this respect. Any experiments or evaluation on this aspect?
> >
> > We thank the reviewer for their question. In our approach, M3GN always predicts the remaining trajectory based on the provided context size. For example, if a trajectory consists of 100 steps and 10 context steps are given, the model predicts the remaining 90 steps, and similarly, with 20 context steps, it predicts the remaining 80 steps. This ensures that the model's output is directly tied to the input context size. However, due to the parameterized nature of the trajectory, the model can handle arbitrary time resolutions during the prediction, offering a distinct advantage over MGN.
> >
> > Our experiments focus on non-periodical tasks, and when different rollout lengths are required, ProDMPs enable smooth transitions between different individual predicted trajectory sections (similar how splines can be connected together)[3].  While this feature is not included in the current paper, we recognize its potential and see it as a valuable direction for future work. Incorporating this capability would enhance the model’s flexibility in handling varying prediction horizons, making it more adaptable across different scenarios.

---

> ### Author Response · Authors · 2024-11-24
> **Rebuttal [3/3]**
>
> > Additionally, splitting single data points into multiple input-output sets seem to increase the effective amount of training data for M3GN, potentially creating an unfair comparison with MGN which use less training data.
>
> We thank the reviewer for their comment. Both M3GN and MGN use the same underlying ground truth trajectories for training. While MGN processes each time step independently, M3GN operates on sequences of varying lengths, reflecting the differences in the training paradigms of the two methods. However, there is no inherent advantage or disadvantage to either approach in terms of the amount of training data used. Furthermore, to ensure a fair comparison, both methods are trained on the same hardware for the same amount of time.
>
> > 9. The authors do not specify how material properties are incorporated. Also, it is unclear whether the test data involve material properties that are in-distribution or out-of-distribution relative to the training data. Providing this information is crucial for evaluating the model's generalization capabilities.
>
> We appreciate the reviewer for pointing out this detail and are happy to provide the requested information. Material properties are explicitly incorporated only in the MGN (Oracle) baseline, as our approach assumes that material properties are unknown and must be inferred from the context. In the Oracle baseline, material properties are added as a global node feature.
>
> For the test splits, all tasks so far involve mainly in-distribution data, with variations in starting positions and collider trajectories, but with material properties that are either identical to or interpolated from those used in training.
>
> To evaluate the model's generalization capabilities, we created a new data split in the Planar Bending task and evaluated all methods on this. Here, the Young's Modulus values used for training ranged between 60 and 500, while the test set included Young's Modulus values from [10, 30, 750, 1000], representing a clearly out-of-distribution scenario. Results from these experiments, included in Figure 5, show that M3GN significantly outperforms both MGN and MGN (Oracle) in this setting, demonstrating strong generalization to out-of-distribution material properties.
>
> We updated the paper to more clearly provide information about the datasets in Appendix D.
>
> > 10. The authors mention that material node features are not added to
> M3GN. Given that these features enhance MGN's performance, it would be
> useful to understand the rationale for this exclusion and perform
> related ablation study.
>
> While additional material information naturally improves prediction performance, it is not often available in realistic scenarios. As an example, fine-tuning a learned dynamics model from sensory data only provides geometry information and sensoric information, but not the internal task properties of the dynamics and the material. As such, M3GN is built around the idea of inferring a latent belief over this information, essentially learning to predict material information from a small context of observed behavior. We thus provided MGN (Material) as an “upper-bound” baseline with perfect knowledge about the material. We thank the reviewer for the valuable comment.
>
> > 11. Although the authors mention other methods in related work besides
> MGN, these methods are not included in the baselines. Some of these
> methods have better accuracy and efficiency. Including these additional
> baselines would provide a clearer view of M3GN’s comparative
> performance.
>
> We extend the related work section and included also more recent graph network simulators. To further strengthen the empirical evaluation, we compared our model against 2 more baselines. The first one is the requested History MGN[1] method, which includes velocities of previous steps as input to the GNN. The second method is the “Equivariant Graph Neural Operator”[4] baseline. We will update the paper with the results, however due to the short time of the rebuttal, this will take time until Wednesday.
>
> > 12. Will the data used in this study be publicly available? Making the
> dataset accessible would facilitate further research and replication
> studies.
>
> We will provide the full codebase and all used datasets upon acceptance of the paper.

---

> > ### Author Response · Authors · 2024-11-24
> > **Rebuttal Concluding Remarks and Sources**
> >
> > We sincerely appreciate the reviewer’s thoughtful and constructive feedback, which has helped improve the quality of our paper. The suggestions provided have been very useful in refining both the methodology and experimental evaluation. We have made extensive improvements, including additional experiments, clarifications, and visualizations, to address the reviewer’s concerns in detail. We believe these refinements significantly enhance the clarity and robustness of our approach.
> > We are hopeful that these revisions, along with the inclusion of further experiments and comparisons, will demonstrate the merits of our work more effectively. We would be grateful if the reviewer would consider these updates in reassessing the manuscript and look forward to any further feedback that could help improve the paper. Thank you again for your time and thoughtful consideration.
> >
> >
> > [1] Pfaff, T., Fortunato, M., Sanchez-Gonzalez, A., & Battaglia, P. Learning Mesh-Based Simulation with Graph Networks. In *International Conference on Learning Representations*.
> >
> > [2] Laurens van der Maaten and Geoffrey Hinton: Visualizing Data using t-SNE  (JMLR, 2008)
> >
> > [3] Li et. al. ProDMP: A Unified Perspective on Dynamic and Probabilistic Movement Primitives (2023, IEEE)
> >
> > [4] Xu et. al. Equivariant Graph Neural Operator for Modeling 3D Dynamics (ICML, 2024)

---

> > > ### Comment · Reviewer_roSh · 2024-12-02
> > > **Reply to Authors**
> > >
> > > I appreciate the authors' efforts to include additional experiments and analyses during the rebuttal period, which address some of my initial questions and concerns regarding this paper. However, many of my original concerns remain, and some of the newly added results lack sufficient discussion on their performance. Specifically, the rationale behind certain modules in the model remains unclear, and the results do not provide enough evidence to justify their inclusion. Below are some of the ongoing issues:
> > >
> > > Q3: The proposed method shows poorer performance at earlier time steps across most tasks (e.g., Figures 15, 16, and 18). Why does this occur? There is no discussion of this phenomenon, making it difficult for readers to understand the advantages and disadvantages of the proposed method.
> > >
> > > Q5: The latent visualization plot is not very informative, as it displays some clusters but also regions where data with different Young's Modulus values are grouped together. What explains this? What do these data points look like, and how do they differ from the rest that clearly form distinct clusters?
> > >
> > > Q6: There is still no statistical information provided regarding memory usage.
> > >
> > > Q7: As previously mentioned, the claim that the scheme is based on meta-learning is inappropriate and may require revisions to the manuscript or the method's description.
> > >
> > > Q8: Although both M3GN and MGN use the same underlying ground truth trajectories, M3GN operates on sequences of varying lengths, which resembles a data augmentation scheme and may thus contribute to M3GN's performance.
> > >
> > > Q11: I appreciate the inclusion of additional comparison methods; however, the EGNO work is only mentioned in the newly updated related work section. In the updated related work, the authors note similarities between EGNO and the proposed work, which were not included in the original draft. This omission is concerning. What is the reason for this? Also, why were the originally mentioned methods in the related work not used for comparison?

---

> > > > ### Author Response · Authors · 2024-12-04
> > > > **Reply to updated questions and concerns**
> > > >
> > > > Thank you for your thoughtful and detailed feedback, which has greatly contributed to improving our manuscript. Below, we provide responses to each point, addressing your concerns and incorporating additional analyses and clarifications where necessary.
> > > >
> > > > >The proposed method shows poorer performance at earlier time steps across most tasks (e.g., Figures 15, 16, and 18). Why does this occur? There is no discussion of this phenomenon, making it difficult for readers to understand the advantages and disadvantages of the proposed method.
> > > >
> > > > We thank the reviewer for pointing out this important aspect of our method’s performance. To address the concern, we investigated this phenomenon primarily in the planar bending task, as only M3GN demonstrates higher errors at earlier time steps. Below, we summarize our findings and outline potential solutions:
> > > >
> > > > 1. **Observed Phenomenon:**
> > > >     - With a context size of 3, the error at earlier time steps disappears. However, with a context size of 2, higher errors occur.
> > > >     - The current explanation is that between steps 2 and 3, the ground truth velocity of certain nodes decreases rapidly (e.g., a significant deceleration). ProDMP uses the velocity at step 2 as a boundary condition. Due to its smoothness constraints, it cannot adjust the velocity rapidly enough, leading to overshooting.
> > > > 2. **Self-Correction:**
> > > >     - Despite the overshooting, the model corrects itself at subsequent steps because it estimates the material properties correctly from the context. While the smoothness constraint limits the initial fit, the context enables the model to recognize and compensate for its earlier inaccuracies.
> > > > 3. **Context Size Dependence:**
> > > >     - When the context size is increased to 3, the rapid velocity change no longer violates the smoothness constraints, resulting in a better fit and elimination of the earlier time step errors.
> > > > 4. **Proposed Solutions:**
> > > >     - **Solution 1:** Introduce more basis functions at the start of the trajectory. By analyzing the statistics of velocity changes in the training data, we can allocate a higher density of basis functions where rapid changes are more likely.
> > > >     - **Solution 2:** Predict the boundary velocity for ProDMP using the context and current velocity at step 2. If a large velocity change is not anticipated, the current velocity can be used directly. This approach could enhance performance without requiring significant architectural modifications.
> > > >
> > > > We plan to train both solutions and include the results in the final version of the paper. Additionally, we will address the impact of smoothness constraints in the discussion of the method’s limitations.
> > > >
> > > > > The latent visualization plot is not very informative, as it displays some clusters but also regions where data with different Young's Modulus values are grouped together. What explains this? What do these data points look like, and how do they differ from the rest that clearly form distinct clusters?
> > > >
> > > > We appreciate the reviewer’s observation regarding the latent visualization plot and its apparent clustering inconsistencies. Upon further investigation, we found the following:
> > > >
> > > > 1. **Latent Variable Behavior:**
> > > >     - The nodes where the latent variables are similar across different material properties exhibit similar predicted trajectories. This suggests that the local latent variable is effectively encoding information about the future trajectory in these cases.
> > > > 2. **Local Encoding Dynamics:**
> > > >     - Since our method employs a local latent variable for each node, it can represent the trajectory explicitly rather than solely clustering based on material properties. This behavior can lead to latent variables that overlap for different materials when their future trajectories align.
> > > > 3. **Planned Discussion:**
> > > >     - We will include a detailed discussion of this phenomenon in the camera-ready version, highlighting how the local latent variable design impacts clustering behavior and trajectory prediction.

---

> > > > > ### Author Response · Authors · 2024-12-04
> > > > > **Reply to updated questions and concerns**
> > > > >
> > > > > > There is still no statistical information provided regarding memory usage.
> > > > >
> > > > > We thank the reviewer for pointing out the need for statistical information on memory usage. To address this, we conducted a GPU memory comparison between MGN, M3GN, and EGNO on three of the most memory-intensive tasks: Deformable Plate, Tissue Manipulation, and Falling Teddy Bear.
> > > > >
> > > > > ### Evaluation Setup:
> > > > >
> > > > > - Memory usage was evaluated for a single prediction with a context size of 10.
> > > > > - Results are summarized in the table below (in MB):
> > > > >
> > > > > | Method | Tissue Manipulation | Deformable Plate | Falling Teddy Bear |
> > > > > | --- | --- | --- | --- |
> > > > > | MGN | 205 MB | 193 MB | 235 MB |
> > > > > | M3GN | 469 MB | 309 MB | 439 MB |
> > > > > | EGNO | 1050 MB | 289 MB | 1600 MB |
> > > > >
> > > > > ### Observations:
> > > > >
> > > > > 1. **Memory Efficiency of M3GN:**
> > > > >     - M3GN's memory usage is approximately double that of MGN for a single prediction.
> > > > >     - However, for tasks with longer prediction horizons, EGNO requires significantly more memory than M3GN. This is because EGNO replicates the entire graph `num_prediction` times and performs message-passing steps over both space and time.
> > > > > 2. **M3GN Design Considerations:**
> > > > >     - In contrast to EGNO, M3GN maintains a single copy of the graph at the anchor step and predicts node trajectories from this representation. This design is more memory-efficient for tasks with extended prediction horizons.
> > > > > 3. **Memory Scalability:**
> > > > >     - Currently, M3GN's memory consumption increases with larger context sizes because we encode the context in parallel. If memory constraints are a concern, this could be easily mitigated by switching to sequential or mini-batch encoding.
> > > > >
> > > > > We hope this information addresses the reviewer’s concern regarding memory usage. We will update the camera ready version with the presented data.
> > > > >
> > > > > > As previously mentioned, the claim that the scheme is based on meta-learning is inappropriate and may require revisions to the manuscript or the method's description.
> > > > >
> > > > > We respectfully disagree with the assertion that our approach does not fall under the category of meta-learning. Below, we clarify our reasoning:
> > > > >
> > > > > 1. **Meta-Learning Framework:**
> > > > >     - A core aspect of meta-learning is the ability to learn from smaller datasets that share a common structure. This is precisely the case for trajectory-level tasks in our method, where the shared structure is the underlying physical dynamics.
> > > > >     - Furthermore, our method employs a meta-learning framework, and we have explicitly explained the mathematical formulation within this context in the manuscript.
> > > > > 2. **Future Directions:**
> > > > >     - As mentioned in the original draft, we acknowledge that learning from other trajectories to infer material properties is an exciting direction for future work. This approach has great potential, and we plan to explore it in future research due to its promising applications.
> > > > >
> > > > > We hope this explanation clarifies our approach and its grounding in meta-learning principles.
> > > > >
> > > > > > Although both M3GN and MGN use the same underlying ground truth trajectories, M3GN operates on sequences of varying lengths, which resembles a data augmentation scheme and may thus contribute to M3GN's performance.
> > > > >
> > > > > We respectfully disagree with the assertion that our method unfairly benefits from a data augmentation scheme. Below, we clarify our reasoning:
> > > > >
> > > > > 1. **Use of Ground Truth Trajectories:**
> > > > >     - Both M3GN and MGN utilize the same underlying ground truth trajectories. However, the way this data is used is an integral part of each method. Since MGN, in the form presented in the baseline paper, is not designed to operate on sequences of varying lengths, we consider this capability a feature of our proposed method.
> > > > >     - It would be a different matter if MGN natively supported sequences of varying lengths, and we failed to implement this. However, this is not the case.
> > > > > 2. **Potential Impact of Data Presentation:**
> > > > >     - While it is possible that this data handling contributes to improved performance even without the meta-learning scheme, this highlights the importance of understanding how different data presentation strategies affect model performance. We believe a benchmark paper exploring such strategies would be valuable for the community.
> > > > > 3. **Fairness of the Comparison:**
> > > > >     - Given that MGN is not inherently capable of handling varying sequence lengths, we do not view this as an unfair advantage but rather a reflection of the strengths of our method.
> > > > >
> > > > > We hope this explanation provides clarity regarding the fairness of our evaluation.

---

> > > > > > ### Author Response · Authors · 2024-12-04
> > > > > > **Reply to updated questions and concerns**
> > > > > >
> > > > > > > I appreciate the inclusion of additional comparison methods; however, the EGNO work is only mentioned in the newly updated related work section. In the updated related work, the authors note similarities between EGNO and the proposed work, which were not included in the original draft. This omission is concerning. What is the reason for this? Also, why were the originally mentioned methods in the related work not used for comparison?
> > > > > >
> > > > > > We thank the reviewer for raising this concern and appreciate the opportunity to clarify.
> > > > > >
> > > > > > 1. **Reason for Omission in the Original Draft:**
> > > > > >     - EGNO is a recent work published at ICML in July 2024. According to the ICLR reviewer guidelines:
> > > > > >
> > > > > >         > "We consider papers contemporaneous if they are published within the last four months. [...] If a paper was published (i.e., at a peer-reviewed venue) on or after July 1, 2024, authors are not required to compare their own work to that paper. [...] Authors are encouraged to cite and discuss all relevant papers, but they may be excused for not knowing about papers not published in peer-reviewed conference proceedings."
> > > > > >         >
> > > > > >     - At the time of submitting the original draft, we were not aware of this paper.
> > > > > > 2. **Inclusion in the Revised Version:**
> > > > > >     - Upon learning about EGNO, we recognized its relevance and included a detailed discussion in the updated related work section. Moreover, we made the additional effort to perform a comparison with EGNO in our experimental evaluation. This demonstrates our commitment to providing a comprehensive and fair assessment of our method in relation to closely related work.
> > > > > > 3. **Regarding Other Methods in Related Work:**
> > > > > >     - Some of the methods, while relevant, are not directly solving the same problem or require slightly different data. For example:
> > > > > >         - Linkerhänger et al.(2023) relies on a stream of point cloud data, which differs from our problem setup.
> > > > > >         - Adaptive meshing strategies, while valuable for larger mesh instances, were not necessary for the tasks we considered. However, this does not imply that our tasks are simple; rather, they emphasize the strengths of our method in handling the given problem scale effectively.
> > > > > >
> > > > > > We hope this clarifies the reasons for the original omission and highlights our efforts to address this in the revised manuscript.
> > > > > >
> > > > > > ## Conclusion
> > > > > > We appreciate the reviewer’s detailed feedback, which has helped us refine our manuscript and analysis. We have addressed all raised concerns to the best of our ability, providing additional experiments, clarifications, and updates where necessary. We believe these additions and revisions substantively improve the manuscript and provide the necessary context for its contributions. We thank the reviewer for their valuable feedback and the time and effort they took to review our work.

---

### Official Review · Reviewer_9DNM · 2024-11-04

**Soundness:** 3
**Presentation:** 3
**Contribution:** 3
**Rating:** 6
**Confidence:** 3

**Summary:**

This paper proposes a graph network simulator for mesh-based simulation on material study. The framework is constructed on a meta-learning problem and applies conditional Neural Processes to address data limitations. This paper shows both qualitative and quantitative experiments.

**Strengths:**

1. This paper shows a clear motivation for initial state uncertainty and data limitation, which are all critical problems in related research fields.

2. Consider the "node-level latent features," which is, to the best of my knowledge, a novel method for solving such a problem.

3. The results of the new simulation task in the paper are convincing for the proposed method.

**Weaknesses:**

1. Some methodology details are unclear, especially in the "Probabilistic Dynamic Movement Primitives" section and "Meta-Learning and Graph Network Simulators."

**Questions:**

1. How does ProDMP generate smooth trajectories based on the predefined conditions of the initial state? Please give detailed justification and explanation.

2. Could the author provide a detailed explanation of how a meta-learning problem can contribute to simulating new scenarios?

---

> ### Author Response · Authors · 2024-11-24
> **Rebuttal**
>
> We thank the reviewer for their thoughtful assessment and for recognizing the strengths of our work. The reviewer has raised important questions regarding the role of ProDMP in generating smooth trajectories and the contribution of the meta-learning framework to simulating new scenarios. Below, we provide detailed explanations to address these points.
>
> ### Responses to Specific Concerns
> > Some methodology details are unclear, especially in the "Probabilistic Dynamic Movement Primitives" section and "Meta-Learning and Graph Network Simulators.”
>
> We appreciate the reviewer's valuable feedback and have made several clarifications and improvements in response. We provide a detailed presentation of the ProDMP approach with its priors works in the appendix. Furthermore, we have reworked and improved the "Meta-Learning and Graph Network Simulators" section to enhance its clarity and ensure a more comprehensive presentation of our methodology.
>
> > How does ProDMP generate smooth trajectories based on the predefined conditions of the initial state? Please give detailed justification and explanation.
>
> We thank the reviewer for the insightful question regarding how ProDMP generates smooth trajectories from predefined initial state conditions. Below, we provide a detailed explanation, the mathematical background can be found in Appendix A of the revised paper.
>
> ProDMPs generate smooth trajectories by extending the concept of DMPs, which are designed to generate smooth motion trajectories for robots or other systems by defining a dynamical system that can be controlled using a set of parameters. The reason this system generates smooth trajectories is that the acceleration and velocity are continuously linked through the ODE. This mathematical formulation prevents sudden jumps in acceleration or velocity, which are what cause jerky, non-smooth motion.
>
> To ensure that the trajectory starts from a specific initial state (i.e., a predefined position and velocity), ProDMPs use a technique to adjust the trajectory parameters such that the trajectory’s position and velocity match the given initial values. These adjustments are made through the introduction of special coefficients, which are computed based on the desired starting position and velocity. This guarantees that the trajectory begins smoothly at the specified initial state.
>
> ProDMPs use a set of mathematical functions, known as basis functions, to define the shape of the trajectory. These functions are predefined and do not change during the learning process, allowing for computational efficiency. The trajectory is built by combining these basis functions with learned parameters that control the trajectory’s final shape. Because the basis functions are continuous and differentiable, they ensure that the trajectory evolves smoothly over time.
>
> >Could the author provide a detailed explanation of how a meta-learning problem can contribute to simulating new scenarios?
>
> Meta-learning, often referred to as "learning to learn," is particularly effective for tasks where a model needs to generalize from one or more previous experiences to adapt to new, previously unseen situations. In the context of simulation, meta-learning enables the model to efficiently leverage past knowledge and quickly adapt to new simulation scenarios.
>
> Meta-learning focuses on teaching models to generalize from prior tasks to new ones by learning shared representations or patterns across various tasks. For simulation, this means the model can learn general dynamics of deformable objects or interactions between objects and environments, rather than needing to learn from scratch for each specific scenario.
>
> The model does not memorize specific tasks but learns a more abstract understanding of how to adapt its behavior to different environments, thus allowing it to simulate a wide range of new scenarios.
>
> In traditional simulation methods, creating accurate models for each new scenario requires substantial computational resources and time. With meta-learning, once the model has learned how to adapt to new contexts, it can perform these adaptations in a more computationally efficient manner, even for complex simulations of deformable objects and their interactions. This could significantly reduce the time and resources needed for new simulation tasks, making it more scalable and efficient.
>
> ### Concluding Remarks
> We hope that the additional details and improvements make our contribution clearer and more accessible. We are grateful for the reviewer’s suggestions, which have helped us strengthen the manuscript, and look forward to any further feedback or questions that may arise.

---

> > ### Author Response · Authors · 2024-12-02
> >
> > We hope this response addresses the reviewer’s concerns. Should the reviewer have any additional questions or require further clarification, we would be happy to provide further details.

---

### Author Response · Authors · 2024-11-28
**Final revision of the paper**

We sincerely thank the reviewers for their thoughtful feedback and detailed questions. Based on these insights, we have made several updates to the paper, including addressing concerns and improving clarity. Below is a summary of the updates made:

### Updates in the Current Rebuttal Version:
1. **New Results and Baselines**:
   - Added results using two new baselines: **MGN** (with current and history node velocities) and the **Equivariant Graph Neural Operator (EGNO)**. These baselines are introduced and evaluated in the updated paper.
   - Modified **M3GN** to include current velocities, improving performance on certain tasks.

2. **Hyperparameter Optimization (HPO)**:
   - Performed HPO on the validation set to determine where MGN and M3GN benefit from velocity information.
   - Reported only the better-performing versions of these methods in the paper.
   - HPO details and results are included in the Appendix.

3. **Visualization and Error Plots**:
   - Updated error plots over time from the previous rebuttal version to include results for the new and updated methods.
   - Main paper qualitative visualizations have been updated to align with the changes in methodology and results.
   - Due to time constraints during the rebuttal phase, qualitative results in Figures 21 to 26 in the Appendix will be updated for the camera-ready version.

### Previous Rebuttal Updates:
1. Enhanced the **Related Work** section by discussing recent Graph Neural Simulation (GNS) methods and Neural Simulation techniques.
2. Added an extensive **ProDMP Appendix** detailing the mathematical background.
3. Rewrote and clarified the sections on **Meta-learning and Graph Network Simulator** and **Model Architecture**.
4. Introduced an **Out-of-Distribution (OOD) Test Dataset** for the Planar Bending task, showcasing generalization ability by testing on material properties outside the training range.
5. Improved **Timing Plots** to compare the runtime of our method and baselines against real simulators.
6. Added M3GN latent space visualization for the Planar Bending task.
7. Extended the Appendix slightly providing more details to the datasets, baselines and hyperparemeteres used.

We hope these changes address the reviewers’ comments comprehensively and demonstrate our efforts to improve the paper based on their valuable feedback.

---

### Meta-Review · Area_Chair_9XwG · 2024-12-22

**Metareview:**

**Summary** The paper propose a Meta-MeshGraphNet model for simulate object deformations. The approach is to meta-learn a graph-network simulator across different types of deformations with varying object materials. For each trajectory, the model is conditioned on the context of initial steps of the simulation that allows to implicitly infer deformation parameters.

**Strengths** The authors demonstrate that graph-network simulators are able to infer deformation parameters from the few initial steps and predict the rest of the simulation for the first time in the literature. They show a variety of deformation problems such as Planar Bending, Deformable Plate, Tissue Manipulation and Teddy Bear Falling.

**Weaknesses** The main weakness and concern is that the paper presents using the history of first few steps of the simulation as the novel approach to meta-learn the deformation parameters, however the previous approaches MeshGraphNet (MGN) and GNS used conditioning on history of several simulation time steps. The reviewers also had questions about rationale behind certain modules, namely ProDMPs. Reviewers point out lack of justification for using ProDMPs and the ablations with and without ProDMPs.

**Decision**  Rejection. Although making the graph network across different deformation parameters is novel, the paper will require substantial reframing to the main claim that MGN and GNS baselines also include the history (Figure 3) and updating the main results (Figures 5 and 6) to include MGN and GNS with history. Therefore, the paper cannot be accepted in its current form.

**Additional Comments On Reviewer Discussion:**

The reviewers had divided views on the paper. Reviewers generally point out that learning and generalizing across deformation parameters is a novel contribution.

However, reviewers brought up two main concerns:
- The justification for using ProDMPs and its generality to various types of simulation is not well described, and it was not sufficiently clarified by the authors during the rebuttal stage.
- As pointed out by reviewer roSh, MeshGraphNet and GNS baselines already included conditioning on the first initial steps and can perform the same task of predicting the simulation on a trajectory with new material properties. During the rebuttal, the authors added the MeshGraphNet results with history in Figure 10. The figure shows that the baselines become on par with the new meta-learning + ProDMPs architecture on Falling Teddy Bear and Planar Bending (OOD) tasks. As a result, the main contribution of the paper in Figure 3 will require substantial revision.

Summary of reviewers’ final evaluations:
- Reviewer roSh: “ the experimental setup and the selection of comparison methods appear to be unfair, making it difficult to clearly demonstrate or justify the advantages of the proposed approach”
- Reviewer iEk9: The novelty of the paper does not meet the standard of a "good paper" for ICLR.

- Reviewer 9DNM This is a technically solid paper on object trajectory and material dynamic generation.
- Reviewer fGhd: This paper is generally solid because it includes new insights into dynamic modeling, especially the trajectory-level meta-learning idea.

---

### Decision · Program_Chairs · 2025-01-22

Reject